# Var-JEPA: A Variational Formulation of the Joint-Embedding Predictive Architecture – Bridging Predictive and Generative Self-Supervised Learning

**Moritz Gögl** [1] **Christopher Yau** [1] [2]

## Abstract

The Joint-Embedding Predictive Architecture (JEPA) is often seen as a non-generative alternative to likelihood-based self-supervised learning, emphasizing prediction in representation space rather than reconstruction in observation space. We argue that the resulting separation from probabilistic generative modeling is largely rhetorical rather than structural: the canonical JEPA design–coupled encoders with a context-to-target predictor–mirrors the variational posteriors and learned conditional priors obtained when variational inference is applied to a particular class of coupled latent-variable models, and standard JEPA can be viewed as a deterministic specialization in which regularization is imposed via architectural and training heuristics rather than an explicit likelihood. Building on this view, we derive the *Variational JEPA* (Var-JEPA), which makes the latent generative structure explicit by optimizing a single Evidence Lower Bound (ELBO). This yields meaningful representations without ad-hoc anti-collapse regularizers and allows principled uncertainty quantification in the latent space. We instantiate the framework for tabular data (Var-T-JEPA) and achieve strong representation learning and downstream performance, improving over T-JEPA across real-world tabular benchmarks while remaining competitive with strong raw-feature baselines.

## 1. Introduction

A main goal in machine learning (ML) is to build models that truly understand and predict the world from data. Self-supervised learning approaches address this challenge by training models to predict one part of an input from another, thereby learning rich representations without requiring human labels. In this context, the Joint-Embedding Predictive Architecture (JEPA) was introduced by LeCun (2022) as a step toward creating a "world model", aiming to learn abstract representations of the world for intelligent decision-making. Unlike regular generative models that predict raw inputs such as pixels (van den Oord et al., 2016b;a; Parmar et al., 2018), the idea behind JEPA is to work entirely in the latent space by predicting the embedding of a target view $y$ from the embedding of a related context signal $x$.

This is implemented using a context encoder $f^{\text{ctx}}$, a target encoder $f^{\text{trg}}$, and a predictor $g$. The model creates a target representation $\hat{s}_y$ from a context representation $s_x$ and an optional auxiliary variable $z$, formulated as:

$$s_x = f^{\text{ctx}}(x), \quad s_y = f^{\text{trg}}(y), \quad \hat{s}_y = g(s_x, z). \quad (1)$$

The training objective minimizes the distance between the predicted and actual target representations, $\mathcal{L}_{\text{JEPA}} = \mathbb{E}\left[\|\hat{s}_y - \text{sg}(s_y)\|_2^2\right]$, where $\text{sg}(\cdot)$ is a stop-gradient operation (Grill et al., 2020). This intentional blocking of gradients into the target encoder is crucial in order to prevent the model from learning trivial predictive features.

While powerful, this design has two consequences. First, it is susceptible to representational collapse, where the encoders learn to output constant vectors, trivially minimizing the loss. This necessitates ad-hoc solutions, such as auxiliary costs $C(\hat{s}_y)$ (Drozdov et al., 2024; Balestriero & LeCun, 2025) or other custom regularization mechanisms such as Exponential Moving Average (EMA) (Assran et al., 2023; Bardes et al., 2024; Thimonier et al., 2025) to enforce representational diversity. Second, JEPA is typically framed as a fundamentally non-generative alternative to likelihood-based self-supervised learning, which suggests a separation from probabilistic latent-variable modeling (LeCun, 2022).

This paper provides a novel perspective on this separation. We argue that it is mostly rhetorical rather than structural: the general JEPA design which includes coupled encoders and a predictor from context to target aligns closely with the variational posteriors and a learned conditional prior in a coupled variational autoencoder (VAE) (Kingma & Welling, 2014; Hao & Shafto, 2023). Concretely, we reverse-engineer a probabilistic model whose Evidence Lower Bound (ELBO) yields the JEPA predictor as a learned latent-space conditional prior, and argue that standard JEPA

[1]University of Oxford, UK. [2]Health Data Research UK. Correspondence to: Moritz Gögl <moritz.gogl@bdi.ox.ac.uk>.

*Proceedings of the 43rd International Conference on Machine Learning*, Seoul, South Korea. PMLR 306, 2026. Copyright 2026 by the author(s).

is naturally interpreted as a deterministic specialization in which regularization is enforced by architectural and training heuristics rather than an explicit likelihood. While the original motivation for JEPA's non-generative framing was to avoid modeling observation noise, our view makes precise when reconstruction is a useful regularizer: it forces latents to encode predictive information while retaining a principled uncertainty semantics. This new framing not only connects predictive and generative self-supervision, but also suggests an inherent route to avoiding collapse and enables latent-space uncertainty quantification.

Specifically, our contributions are: (1) We establish a novel formal link between JEPA and variational inference, reframing it as a deterministic latent variable model. (2) We introduce Var-JEPA, a generative JEPA derived from this link, with a corresponding ELBO objective. (3) We show how this ELBO naturally prevents representational collapse without needing the surrogate losses common in JEPA and provide a connection to the recently proposed SIGReg regularization in LeJEPA (Balestriero & LeCun, 2025). (4) We develop a practical Var-T-JEPA implementation for heterogeneous tabular data. (5) We empirically validate our approach in a controlled simulation study and across multiple downstream datasets, showing gains over standard JEPA on real-world tabular benchmarks while remaining competitive with strong baselines trained on raw features.

## 2. Related Work

**JEPA-style predictive representation learning.** JEPA was proposed as a general framework for learning predictive world models by matching representations (LeCun, 2022), an idea with roots in earlier unsupervised representation learning (Schmidhuber & Prelinger, 1993). Notable implementations include I-JEPA (Assran et al., 2023), which demonstrates that this approach learns strong image representations without handcrafted data augmentations by predicting embeddings of masked regions from context regions. V-JEPA extends the approach to video, learning spatiotemporal representations that support understanding and prediction (Bardes et al., 2024). T-JEPA (Thimonier et al., 2025) adapts JEPA to feature-level masking and transformer tokenization for heterogeneous tabular data, showing competitive tabular representation learning without the heavy reliance on typical data augmentations.

**Avoiding collapse via distributional regularization.** Common JEPA implementations use EMA to stabilize training by slowly updating the target encoder (Assran et al., 2023; Bardes et al., 2024), but this heuristic approach does not directly control the distributional properties of the learned embeddings. In contrast, Drozdov et al. (2024) prevent collapse by explicitly regularizing embedding distributions via variance–covariance penalties that encourage

feature-wise variance and discourage correlated dimensions. LeJEPA (Balestriero & LeCun, 2025) provides a complementary view: it argues that an (approximately) isotropic Gaussian embedding distribution is minimax-favorable for downstream probing under broad probe families, and it proposes Sketched Isotropic Gaussian Regularization (SIGReg) as a scalable way to match the aggregated embedding distribution to $\mathcal{N}(0, I)$ via random one-dimensional projections. Formally, SIGReg applies a univariate statistical test $T$ (typically the Epps-Pulley test (Epps & Pulley, 1983)) to projections of embeddings along random unit-norm directions $\mathbb{A} = \{a_1, \ldots, a_M\}$:

$$\text{SIGReg}(\mathbb{A}, \{e_n\}_{n=1}^N) = \frac{1}{|\mathbb{A}|} \sum_{a \in \mathbb{A}} T(\{a^\top e_n\}_{n=1}^N), \quad (2)$$

where $\{e_n\}_{n=1}^N$ denotes a batch of embeddings and $T$ measures the discrepancy between the empirical distribution of projected embeddings and the standard normal distribution. In our work, the ELBO provides per-sample regularization toward fixed priors for $s_x$ and $z$, while we study SIGReg as an explicit aggregated-distribution regularizer (especially for $s_y$) in the simulation study.

**Generative world models and latent-space modeling.** World-model research has emphasized learning latent dynamics for prediction and planning (Ha & Schmidhuber, 2018). Compared to generative latent models and VAEs (Kingma & Welling, 2014; Bowman et al., 2016), JEPA emphasizes prediction in representation space and often avoids explicit likelihood modeling. Our main contribution is to show that a JEPA-like architecture arises naturally from a variational latent-variable model with a learned conditional prior. While concurrent work by Huang (2026) also explores a probabilistic formulation of JEPA for uncertainty-aware latent prediction, Var-JEPA instead formulates JEPA as a coupled latent-variable generative model with a unified ELBO, thereby bridging predictive joint-embedding learning and generative modeling in a single objective. This shows that Var-JEPA is not an incremental regularizer over JEPA, but a rigorous variational formulation that exposes the latent generative structure implicit in the predictor-encoder pattern.

## 3. The Variational Perspective on JEPA
### 3.1. Problem Formulation
We begin by asking a simple structural question: *if the predictive embedding steps in JEPA were interpreted as variational posteriors of a coupled VAE, what generative model would give rise to those posteriors?*

This viewpoint naturally leads us to a novel reinterpretation of JEPA within a probabilistic latent-variable framework. Concretely, we replace JEPA's deterministic encoders and predictor with conditional distributions and interpret the predictive pathway as a learned latent-space conditional prior.

This creates a clear generative process over context, target, and auxiliary latent variables, from which a single variational objective (ELBO) emerges. Our formulation unifies JEPA-style predictive learning with reconstruction and conditional generation, while providing a rigorous mechanism for avoiding collapse through latent regularization.

**Structure.** Like JEPA, our model operates on context observations $x \in \mathbb{R}^D$ and target observations $y \in \mathbb{R}^D$, learning latent representations $s_x \in \mathbb{R}^d$ (context), $s_y \in \mathbb{R}^d$ (target), and $z \in \mathbb{R}^{d_z}$ (auxiliary predictive variable to capture variability in $s_y$ that $s_x$ cannot explain). The directed acyclic graph (DAG) below makes the assumed conditional dependencies among observations and latents explicit, and follows the underlying generative process:

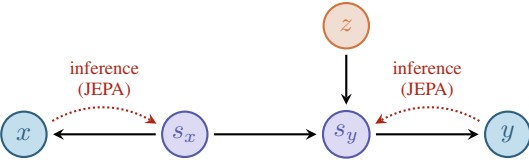

While a standard JEPA primarily operates with unidirectional mappings from observations to representations ($x \rightarrow s_x, y \rightarrow s_y$), our variational framework requires bidirectional relationships to model both the generative and inference processes. For both the context and target, we learn two directions: encoding (from observations to latents) and reconstruction (from latents back to observations). A single ELBO objective ties these directions together and trains encoders and decoders jointly.

**Objective and factorization.** Our main objective is to learn a generative model that maximizes the marginal log-likelihood of observed data pairs $(x, y)$:

$$\max_\theta \mathbb{E}_{(x,y)\sim\text{data}} \left[\log p_\theta(x, y)\right] = \tag{3}$$

$$\max_\theta \mathbb{E}_{(x,y)\sim\text{data}} \left[\log \iiint p_\theta(x, y, s_x, z, s_y) ds_x \, dz \, ds_y\right].$$

The log-likelihood $\log p_\theta(x, y)$ measures how well our parameterized model (with parameters $\theta$) explains the observed data. Following the DAG, the joint distribution $p_\theta(x, y, s_x, z, s_y)$ factorizes as

$$p_\theta(x, y, s_x, z, s_y) = p(s_x) \cdot p(z) \cdot p_\theta(x \mid s_x) \tag{4}$$
$$\cdot p_\theta(s_y \mid s_x, z) \cdot p_\theta(y \mid s_y).$$

**Generative model parameterization.** We implement distributions in the generative model as Gaussians. The priors over the latent variables $s_x$ and $z$ are standard Gaussians, while the remaining conditional distributions are parameterized by neural networks:

$$p(s_x) = \mathcal{N}(s_x; 0, I), \tag{5a}$$

$$p(z) = \mathcal{N}(z; 0, I), \tag{5b}$$

$$p_\theta(x \mid s_x) = \mathcal{N}(x; U_\theta^x(s_x), \sigma_x^2 I), \tag{5c}$$

$$p_\theta(s_y \mid s_x, z) = \mathcal{N}(s_y; \mu_\theta^{s_y}(s_x, z), \Sigma_\theta^{s_y}(s_x, z)), \tag{5d}$$

$$p_\theta(y \mid s_y) = \mathcal{N}(y; U_\theta^y(s_y), \sigma_y^2 I). \tag{5e}$$

The decoder networks $U_\theta^x : \mathbb{R}^{D_{\text{hid}}} \rightarrow \mathbb{R}^{D_{\text{obs}}}$ and $U_\theta^y : \mathbb{R}^{D_{\text{hid}}} \rightarrow \mathbb{R}^{D_{\text{obs}}}$ reconstruct observations from their latent representations, where $D_{\text{obs}}$ represents the observation dimension. The predictive network outputs $\mu_\theta^{s_y}(s_x, z)$ and $\Sigma_\theta^{s_y}(s_x, z)$ are computed by neural networks with parameters $\theta$, generating distributional parameters for the target latent $s_y$ conditioned on context $s_x$ and auxiliary variable $z$. The reconstruction noise parameters $\sigma_x^2$ and $\sigma_y^2$ may be set to 1 or learned globally to model observation uncertainty.

### 3.2. Variational Posterior

Since the integral in Eq. (3) is intractable under the given factorization and parameterization, due to complex neural network functions and high-dimensional latent spaces, we cannot optimize $\log p_\theta(x, y)$ directly. Instead, we employ a variational inference approach by introducing a tractable variational posterior $q_\phi(s_x, z, s_y \mid x, y)$ as an approximation of the true posterior $p(s_x, z, s_y \mid x, y)$, and parameterized by neural networks with parameters $\phi$. We factorize the variational posterior as:

$$q_\phi(s_x, z, s_y \mid x, y) = q_\phi(s_x \mid x) \cdot q_\phi(z \mid s_x)$$
$$\cdot q_\phi(s_y \mid s_x, z, y). \tag{6}$$

We adopt this factorization for the following reasons:

- The context latent $q_\phi(s_x \mid x)$ depends only on the context observation $x$, ensuring that context representations are learned independently of target information.

- The auxiliary latent $z$ depends only on the context representation $s_x$ to prevent information leakage from targets during training.

- The target posterior $q_\phi(s_y \mid s_x, z, y)$ depends on both $s_x$ and $z$ as well as the target observation $y$, as we will use a reconstruction term to regularize the learning of $s_y$ to encode meaningful target information, ensuring that the dependence on context is learned through proper predictive relationships rather than trivial shortcuts.

**Variational posterior parameterization.** Each component of the variational posterior is parameterized as a Gaussian distribution with learnable mean and covariance:

$$q_\phi(s_x \mid x) = \mathcal{N}(s_x; \mu_\phi^{s_x}(x), \Sigma_\phi^{s_x}(x)), \tag{7a}$$

$$q_\phi(z \mid s_x) = \mathcal{N}(z; \mu_\phi^z(s_x), \Sigma_\phi^z(s_x)), \tag{7b}$$

$$q_\phi(s_y \mid s_x, z, y) = \mathcal{N}(s_y; \mu_\phi^{s_y}(s_x, z, y), \Sigma_\phi^{s_y}(s_x, z, y)). \tag{7c}$$

The inference networks $\mu_\phi^{s_x}$, $\Sigma_\phi^{s_x}$, $\mu_\phi^z$, $\Sigma_\phi^z$, $\mu_\phi^{s_y}$, and $\Sigma_\phi^{s_y}$ are implemented as neural networks that output the distributional parameters given their respective inputs.

### 3.3. Var-JEPA: Evidence Lower Bound

Having established the variational posterior, we derive the ELBO, a computable variational lower bound on the marginal log-likelihood via Jensen's inequality:

$$
\begin{aligned}
\log p_\theta(x,y) \geq \mathbb{E}_{q_\phi}\big[&\log p_\theta(x,y,s_x,z,s_y)- \\
&\log q_\phi(s_x,z,s_y \mid x,y)\big] \equiv \text{ELBO}(x,y;\theta,\phi).
\end{aligned}
\tag{8}
$$

More specifically, we achieve our goal of maximizing $\log p_\theta(x,y)$ by maximizing its tractable lower bound. Substituting the factorizations of $p_\theta$ and $q_\phi$, we derive:

$$
\begin{aligned}
&\text{ELBO}(x,y;\theta,\phi) = \\
&= \mathbb{E}_{q_\phi}\big[\log p(s_x)+\log p(z)+\log p_\theta(x\mid s_x)+\log p_\theta(s_y|s_x,z)+ \\
&\quad \log p_\theta(y|s_y)-\log q_\phi(s_x|x)-\log q_\phi(z|s_x)-\log q_\phi(s_y|s_x,z,y)\big] \\
&= \mathbb{E}_{q_\phi(s_x|x)}\big[\log p_\theta(x|s_x)\big]+\mathbb{E}_{q_\phi(s_x,z,s_y|x,y)}\big[\log p_\theta(y\mid s_y)\big]+ \\
&\quad \mathbb{E}_{q_\phi(s_x|x)}\big[\log p(s_x)-\log q_\phi(s_x|x)\big]+\mathbb{E}_{q_\phi(s_x,z|x)}\big[\log p(z)- \\
&\quad \log q_\phi(z\mid s_x)\big]+\underbrace{\mathbb{E}_{q_\phi(s_x,z,s_y|x,y)}\big[\log p_\theta(s_y\mid s_x,z)\big]}_{\text{(a)}}+ \\
&\quad \underbrace{\mathbb{E}_{q_\phi(s_x,z,s_y|x,y)}\big[-\log q_\phi(s_y\mid s_x,z,y)\big]}_{\text{(b)}}.
\end{aligned}
$$

We recognize that $\mathbb{E}_{q_\phi}[\log p(\cdot)-\log q_\phi(\cdot)]=-\text{KL}(q_\phi\|p)$ and that the last two terms (a) and (b) can be combined into a single KL divergence term: $-\text{KL}\big(q_\phi(s_y|s_x,z,y)\|p_\theta(s_y|s_x,z)\big)$. This gives us the final expression for the ELBO:

$$
\begin{aligned}
\text{ELBO}(x,y;\theta,\phi) = &\ \mathbb{E}_{q_\phi(s_x|x)}\big[\log p_\theta(x\mid s_x)\big]+ \\
&\mathbb{E}_{q_\phi(s_x,s_y|x,y)}\big[\log p_\theta(y|s_y)\big]-\text{KL}\big(q_\phi(s_x\mid x)\|p(s_x)\big)- \\
&\text{KL}\big(q_\phi(z|s_x)\|p(z)\big)-\text{KL}\big(q_\phi(s_y|s_x,z,y)\|p_\theta(s_y\mid s_x,z)\big)
\end{aligned}
\tag{9}
$$

This objective can be viewed as a coupled VAE: $q_\phi(s_x \mid x)$, $q_\phi(z \mid s_x)$, and $q_\phi(s_y \mid s_x,z,y)$ are the amortized posteriors; $p_\theta(x \mid s_x)$ and $p_\theta(y \mid s_y)$ are the decoders; and $p_\theta(s_y \mid s_x,z)$ acts as a conditional prior that couples the two latent spaces, recovering the JEPA predictor as a generative component.

During training, our objective is to maximize the ELBO: $\max_{\phi,\theta}\ \mathbb{E}_{(x,y)\sim\text{data}}\big[\text{ELBO}(x,y;\theta,\phi)\big]$. In practice, we convert this to a minimization problem by defining a loss function as a weighted combination of the negative ELBO terms. Specifically, we minimize $\mathcal{L}_{\text{ELBO}}(x,y;\theta,\phi)$, which decomposes the ELBO into 5 interpretable terms, each with its own scalar weight:

$$
\begin{aligned}
\mathcal{L}_{\text{ELBO}}(x,y;\theta,\phi) = &\ \alpha^{\text{rec}}\Big(\underbrace{-\mathbb{E}_{q_\phi(s_x|x)}\big[\log p_\theta(x\mid s_x)\big]}_{\mathcal{L}^{\text{rec}}}\Big) \\
&+ \alpha^{\text{gen}}\Big(\underbrace{-\mathbb{E}_{q_\phi(s_x,z,s_y|x,y)}\big[\log p_\theta(y\mid s_y)\big]}_{\mathcal{L}^{\text{gen}}}\Big) \\
&+ \alpha_{s_x}^{\text{KL}}\underbrace{\text{KL}\big(q_\phi(s_x\mid x)\|p(s_x)\big)}_{\mathcal{L}_{s_x}^{\text{KL}}}+\alpha_z^{\text{KL}}\underbrace{\text{KL}\big(q_\phi(z\mid s_x)\|p(z)\big)}_{\mathcal{L}_z^{\text{KL}}} \\
&+ \alpha_{s_y}^{\text{KL}}\underbrace{\text{KL}\big(q_\phi(s_y\mid s_x,z,y)\|p_\theta(s_y\mid s_x,z)\big)}_{\mathcal{L}_{s_y}^{\text{KL}}\equiv-[(a)+(b)]}
\end{aligned}
\tag{10}
$$

The loss terms have the following practical interpretation:

$\mathcal{L}^{\text{rec}}$ *(Context Reconstruction)*: Measures the reconstruction quality–how accurately we can recover the original context observation $x$ from its latent representation $s_x$.

$\mathcal{L}^{\text{gen}}$ *(Target Generation)*: Measures the generation quality–how accurately the target latent $s_y$ can reconstruct the actual target observation $y$.

$\mathcal{L}_{s_x}^{\text{KL}}$ *(KL on $s_x$)*: Regularizes the context latent distribution $s_x$ toward the prior $\mathcal{N}(0,I)$, maintaining a well-behaved latent space for $s_x$.

$\mathcal{L}_z^{\text{KL}}$ *(KL on $z$)*: Regularizes the auxiliary latent distribution $z$ toward the prior $\mathcal{N}(0,I)$, ensuring $z$ doesn't become overly complex or specialized.

$\mathcal{L}_{s_y}^{\text{KL}}$ *(KL on $s_y$)*: Regularizes the target latent posterior toward the generative model prediction.

   (a) *(Prediction)*: Measures the predictive accuracy–how well $s_x$ and auxiliary variable $z$ can jointly forecast the target latent representation $s_y$.

   (b) *(Entropy)*: Maintains distributional diversity of the target posterior $q_\phi(s_y\mid s_x,z,y)$ by encouraging it to remain sufficiently uncertain, therefore preventing collapse.

### 3.4. JEPA vs. Var-JEPA Architectures

Fig. 1 illustrates the relationship between standard JEPA and our Var-JEPA. Both approaches share the same core prediction structure with context ($x$) and target ($y$) observations, their latent representations $s_x, s_y$, and auxiliary latent variable $z$. JEPA relies on inference and prediction networks, which requires surrogate costs $C(\cdot)$ to prevent the collapse of representations. Var-JEPA adds generative networks (i.e., decoders) so the model can be trained with a variational objective which naturally prevents collapse.

### 3.5. Reparameterization Trick and Sampling Implementation

To backpropagate through stochastic latents, we apply the reparameterization trick (Kingma & Welling, 2014), express-

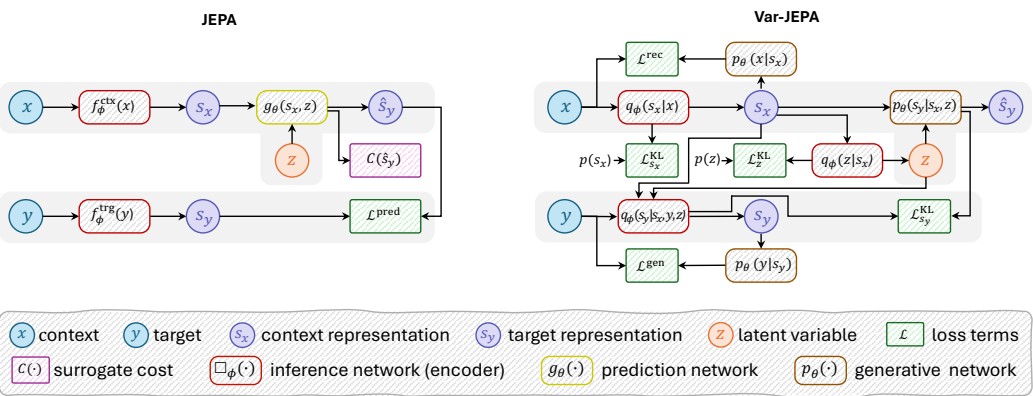

Figure 1. Comparison between JEPA (left) and Var-JEPA (right). Var-JEPA extends JEPA by replacing deterministic encoders and predictor with probabilistic distributions, and adding generative networks (decoders) to enable variational learning under a unified ELBO.

ing samples as deterministic functions of noise so gradients are well-defined. The latent variables are sampled using the following reparameterized forms:

$$s_x = \mu_\phi^{s_x}(x) + \left[\Sigma_\phi^{s_x}(x)\right]^{1/2} \epsilon_{s_x},$$
$$z = \mu_\phi^z(s_x) + \left[\Sigma_\phi^z(s_x)\right]^{1/2} \epsilon_z,$$
$$s_y = \mu_\phi^{s_y}(s_x, z, y) + \left[\Sigma_\phi^{s_y}(s_x, z, y)\right]^{1/2} \epsilon_{s_y},$$

where $\epsilon_{s_x}, \epsilon_z, \epsilon_{s_y} \sim \mathcal{N}(0, I)$ are independent standard Gaussian noise vectors, and $[\Sigma_\phi^{(\cdot)}(\cdot)]^{1/2}$ denotes the matrix square root of the covariance. In practice, we estimate the gradients by sampling a single realization of each latent variable per forward pass. The gradient estimator becomes:

$$\nabla_{\theta,\phi} \mathcal{L} \approx \nabla_{\theta,\phi}\big[\log p_\theta(x,y,s_x,z,s_y) - \log q_\phi(s_x,z,s_y|x,y)\big],$$

where each training step uses new samples $s_x, z, s_y$ drawn from the reparameterized distributions.

### 3.6. Theoretical Generative Inference

When Var-JEPA is interpreted as a truly generative model, it can generate target observations from context by sampling through the generative pathway. The generative inference procedure, applicable when targets are unavailable, is:

$$s_x \sim q_\phi(s_x \mid x) = \mathcal{N}\big(s_x; \mu_\phi^{s_x}(x), \Sigma_\phi^{s_x}(x)\big),$$
$$z \sim p(z) = \mathcal{N}(z; 0, I),$$
$$s_y \sim p_\theta(s_y \mid s_x, z) = \mathcal{N}\big(s_y; \mu_\theta^{s_y}(s_x, z), \Sigma_\theta^{s_y}(s_x, z)\big),$$
$$y \sim p_\theta(y \mid s_y) = \mathcal{N}\big(y; U_\theta^y(s_y), \sigma_y^2 I\big).$$

This differs from training, where $z$ is inferred via $q_\phi(z \mid s_x)$ and the target posterior $q_\phi(s_y \mid s_x, z, y)$ has access to ground truth targets. For downstream representation learning tasks, where the goal is predictive rather than generative, we use deterministic embeddings by reporting posterior means rather than sampling (see Appendix D.5).

### 3.7. Relationship to LeJEPA

LeJEPA (Balestriero & LeCun, 2025) motivates isotropic Gaussian embeddings as minimax-optimal for downstream prediction under a broad class of probes. It enforces this distributional structure using SIGReg, which matches the *aggregated* embedding distribution to an isotropic Gaussian via random one-dimensional projections.

Var-JEPA relates to this picture through its variational regularization terms. For latent variables with a fixed standard-normal prior ($s_x$ and $z$), the ELBO contains per-sample KL terms $\mathbb{E}\,\mathrm{KL}(q_\phi(\cdot) \,\|\, \mathcal{N}(0, I))$. These admit the standard "ELBO surgery" decomposition (Hoffman & Johnson, 2016) into an aggregated posterior mismatch and an information-bottleneck term:

$$\mathbb{E}_x \mathrm{KL}\big(q_\phi(s_x \mid x) \,\|\, \mathcal{N}(0, I)\big) = \mathrm{KL}\big(q_\phi(s_x) \,\|\, \mathcal{N}(0, I)\big) + I_{q_\phi}(x; s_x), \tag{11}$$

with an analogous identity for $z$. In contrast, the target-latent term in Var-JEPA is $\mathrm{KL}(q_\phi(s_y \mid s_x, z, y) \,\|\, p_\theta(s_y \mid s_x, z))$, i.e., regularization toward a *learned conditional prior* rather than $\mathcal{N}(0, I)$, and therefore does not decompose into an aggregated KL to a fixed reference distribution in the same way. This motivates studying SIGReg as an explicit aggregated-distribution regularizer, particularly for $s_y$, and reporting LeJEPA-link diagnostics in the simulation study.

## 4. Experiments

### 4.1. Simulation Study

We evaluate Var-JEPA in a controlled synthetic setting designed to isolate the effects of latent regularization, isotropy, and information bottlenecks while retaining a known ground-truth generative process. This setting allows us to analyze how the variational objective shapes the aggregated latent distribution and how this, in turn, affects downstream probe performance.

**Data generation.** We generate observations $(x_i, y_i)$ from a latent process with a mixture-structured context latent $s_{x_i}$, a target latent $s_{y_i}$ that is correlated with $s_{x_i}$, and an auxiliary factor $z_i$ that influences $s_{y_i}$ and therefore $y_i$. Observations are obtained by passing these latents through nonlinear maps $h_x$ and $h_y$ and adding independent Gaussian noise:

$$z_i \sim \mathcal{N}(0, \Sigma_z), \quad s_{x_i} \sim \tfrac{1}{2}\mathcal{N}(0, \Sigma_x) + \tfrac{1}{2}\mathcal{N}(\delta, \Sigma_x),$$
$$s_{y_i} \mid s_{x_i}, z_i \sim \mathcal{N}(s_{x_i} + A z_i, \Sigma_y),$$
$$x_i = h_x(s_{x_i}) + \epsilon_x, \ \epsilon_x \sim \mathcal{N}(0, \tau_x^2),$$
$$y_i = h_y(s_{y_i}) + \epsilon_y, \ \epsilon_y \sim \mathcal{N}(0, \tau_y^2).$$

**ELBO decomposition and interpretation.** Although Var-JEPA does not explicitly optimize aggregated-posterior KL terms, the expected KL terms for latents with fixed priors (here $s_x$ and $z$) admit the standard decomposition in Eq. (11). For the target latent, the ELBO instead contains a conditional-prior term that does not decompose into an aggregated KL to a fixed reference distribution in the same way. This highlights that Var-JEPA couples distributional regularization of latent spaces *with fixed priors* with information bottlenecks that penalize excessive dependence between inputs and latents.

**Experimental variants.** To study the relationship between Var-JEPA and LeJEPA, we perform a systematic ablation study comparing ten variants (labeled A–J in Table 1). Unless explicitly set to zero, all loss weight parameters use default values: $\alpha^{\text{rec}} = \alpha^{\text{gen}} = \alpha^{\text{KL}}_{s_x} = \alpha^{\text{KL}}_z = \alpha^{\text{KL}}_{s_y} = 1$ for ELBO terms, and $\lambda_{s_x} = \lambda_{s_y} = 10$ for SIGReg regularization.

We compare the following variants: (A) the full ELBO objective (standard Var-JEPA); (B–D) ELBO augmented with SIGReg, where SIGReg is applied to $s_x$ only (B, $\lambda_{s_y} = 0$), $s_y$ only (C, $\lambda_{s_x} = 0$), or both (D). The general form of ELBO+SIGReg is:

$$\mathcal{L}_{\text{ELBO+SIGReg}} = \mathcal{L}_{\text{ELBO}} + \lambda_{s_x}\text{SIGReg}(\{s_{x,i}\}_{i=1}^{B}, \mathcal{N}(0, I))$$
$$+ \lambda_{s_y}\text{SIGReg}(\{s_{y,i}\}_{i=1}^{B}, \mathcal{N}(0, I)).$$

We also consider (E–F) ELBO with some KL terms removed ($\alpha^{\text{KL}}_{s_x} = 0$ and $\alpha^{\text{KL}}_{s_y} = 0$, respectively); (G) ELBO with reconstruction and generation terms removed ($\alpha^{\text{rec}} = \alpha^{\text{gen}} = 0$), leaving only KL regularization; (H) the same as (G) but with SIGReg added; (I) ELBO with all KL terms removed ($\alpha^{\text{KL}}_{s_x} = \alpha^{\text{KL}}_z = \alpha^{\text{KL}}_{s_y} = 0$), leaving only reconstruction and generation; and (J) the same as (I) but with SIGReg added, which corresponds to setting all KL terms to zero in $\mathcal{L}_{\text{ELBO+SIGReg}}$, resulting in a surrogate objective where per-sample KL terms are replaced by distribution-level SIGReg regularization. These variants allow us to disentangle the effects of per-sample variational regularization from direct control of the aggregated latent distributions.

**Simulation study results.** We evaluate a controlled comparison with identical architecture and training schedule, varying only the presence/scope of SIGReg and whether KL terms are included. Results are shown in Table 1, which reports distribution diagnostics: aggregated KL divergence to $\mathcal{N}(0, I)$, SIGReg-MSE (discrepancy via SIGReg), Frobenius norm of covariance deviation from identity $\|\text{Cov}(s) - I\|_F$, and mean norm $\|\mathbb{E}[s]\|_2$. Figure 2 shows how these diagnostics evolve during training for selected ablations.

The key finding is that the KL divergence terms in the ELBO achieve distributional properties for $s_x$ that are comparable to those obtained with explicit aggregated-distribution regularization via SIGReg. This demonstrates that per-sample KL regularization toward fixed priors naturally enforces aggregated-distribution isotropy, without requiring additional regularization mechanisms. For $s_y$, the ELBO regularizes toward a learned conditional prior rather than $\mathcal{N}(0, I)$, which is the theoretically correct objective for the target latent; consequently, the aggregated distribution deviates from isotropic Gaussian, as expected.

Removing reconstruction terms (variant G) causes representational collapse, evidenced by probe accuracy dropping to near chance-level. Removing KL terms on $s_x$ (variant E) leads to severe distributional collapse, with aggregated KL divergences and SIGReg-MSE values increasing dramatically. Removing all KL terms (variant I) causes even more severe collapse across all distributional metrics, demonstrating that KL regularization is essential for maintaining well-behaved latent distributions. When SIGReg is added to the no-KL variant (variant J), it partially compensates for the missing KL terms, but the full ELBO remains optimal. The training dynamics in Figure 2 reveal that these distributional properties emerge gradually during training, with KL terms providing stable regularization throughout.

### 4.2. Downstream Evaluation on Tabular Data
#### 4.2.1. EXPERIMENTAL DETAILS
**Var-T-JEPA.** We evaluate our tabular implementation, Var-T-JEPA, which combines feature-level masking with the unified variational objective from Eq. (10). The model is inspired by the deterministic T-JEPA (Thimonier et al., 2025), but instantiates the Var-JEPA framework for tabular data by learning Gaussian latent embeddings and training the prediction and reconstruction pathways jointly under the ELBO. Var-T-JEPA tokenizes heterogeneous numerical and categorical features into a transformer sequence, infers Gaussian latent embeddings $s_x$ and $s_y$, and trains a latent-space predictor via a coupled reconstruction–prediction objective that directly instantiates the Var-JEPA framework for heterogeneous tabular data. This yields both deterministic embeddings (via posterior means) and per-sample uncertainty estimates from the learned latent distributions. We defer a full conceptual description to Appendix A.

*Table 1.* Simulation study: Var-JEPA and SIGReg ablations (mean $\pm$ std over 5 runs). We report linear-probe accuracy (predicting the mixture component) and LeJEPA-link diagnostics for the aggregated latent distributions of $s_x$ and $s_y$. For $s_y$ we additionally report the mean conditional-prior coupling term $\mathbb{E}\,\mathrm{KL}(q_\phi(s_y\,|\,s_x, z, y)\,\|\,p_\theta(s_y\,|\,s_x, z))$.

| Exp. | Objective | Acc($s$) | $\mathrm{KL}_{\mathrm{agg}}(q(s)\|\mathcal{N}(0,I))$ | SIGReg-MSE($s$) | $\|\mathrm{Cov}(s) - I\|_F$ | $\|\mathbb{E}[s]\|_2$ |
|---|---|---|---|---|---|---|
| | | | **Context latent diagnostics ($s_x$)** | | | |
| (A) | ELBO (Var-JEPA) | $0.996 \pm 0.002$ | $0.113 \pm 0.037$ | $4.0\mathrm{e}{-4} \pm 1.0\mathrm{e}{-4}$ | $0.649 \pm 0.127$ | $0.082 \pm 0.009$ |
| (B) | ELBO+SIGReg ($\lambda_{s_x}=10, \lambda_{s_y}=0$) | $0.996 \pm 0.002$ | $0.108 \pm 0.032$ | $4.0\mathrm{e}{-4} \pm 1.0\mathrm{e}{-4}$ | $0.639 \pm 0.113$ | $0.077 \pm 0.018$ |
| (C) | ELBO+SIGReg ($\lambda_{s_x}=0, \lambda_{s_y}=10$) | $0.996 \pm 0.002$ | $0.110 \pm 0.033$ | $4.0\mathrm{e}{-4} \pm 1.0\mathrm{e}{-4}$ | $0.647 \pm 0.118$ | $0.077 \pm 0.018$ |
| (D) | ELBO+SIGReg | $0.996 \pm 0.002$ | $0.107 \pm 0.030$ | $3.9\mathrm{e}{-4} \pm 1.0\mathrm{e}{-4}$ | $0.634 \pm 0.108$ | $0.078 \pm 0.018$ |
| (E) | ELBO ($\alpha^{\mathrm{KL}}_{s_x}=0$) | $0.996 \pm 0.002$ | $8.374 \pm 1.944$ | $5.1\mathrm{e}{-2} \pm 1.9\mathrm{e}{-2}$ | $7.197 \pm 1.522$ | $2.550 \pm 0.607$ |
| (F) | ELBO ($\alpha^{\mathrm{KL}}_{s_y}=0$) | $0.996 \pm 0.002$ | $0.098 \pm 0.038$ | $3.0\mathrm{e}{-4} \pm 1.0\mathrm{e}{-4}$ | $0.597 \pm 0.124$ | $0.083 \pm 0.010$ |
| (G) | ELBO ($\alpha^{\mathrm{rec}}=\alpha^{\mathrm{gen}}=0$) | $0.571 \pm 0.010$ | $0.015 \pm 0.002$ | $1.6\mathrm{e}{-4} \pm 1.1\mathrm{e}{-5}$ | $0.229 \pm 0.016$ | $0.055 \pm 0.005$ |
| (H) | ELBO ($\alpha^{\mathrm{rec}}=\alpha^{\mathrm{gen}}=0$)+SIGReg | $0.834 \pm 0.044$ | $0.015 \pm 0.001$ | $1.7\mathrm{e}{-4} \pm 1.4\mathrm{e}{-5}$ | $0.225 \pm 0.008$ | $0.065 \pm 0.006$ |
| (I) | ELBO ($\alpha^{\mathrm{KL}}_{s_x}=\alpha^{\mathrm{KL}}_{z}=\alpha^{\mathrm{KL}}_{s_y}=0$) | $0.995 \pm 0.002$ | $10.465 \pm 2.207$ | $5.7\mathrm{e}{-2} \pm 1.9\mathrm{e}{-2}$ | $8.807 \pm 1.208$ | $3.100 \pm 0.594$ |
| (J) | ELBO ($\alpha^{\mathrm{KL}}_{s_x}=\alpha^{\mathrm{KL}}_{z}=\alpha^{\mathrm{KL}}_{s_y}=0$) + SIGReg | $0.996 \pm 0.002$ | $2.115 \pm 0.093$ | $3.3\mathrm{e}{-3} \pm 5.4\mathrm{e}{-4}$ | $2.594 \pm 0.077$ | $0.205 \pm 0.032$ |
| | | | **Target latent diagnostics ($s_y$)** | | | |
| (A) | ELBO (Var-JEPA) | $0.993 \pm 0.001$ | $3.530 \pm 0.377$ | $1.7\mathrm{e}{-2} \pm 1.7\mathrm{e}{-3}$ | $4.727 \pm 0.528$ | $1.320 \pm 0.180$ |
| (B) | ELBO+SIGReg ($\lambda_{s_x}=10, \lambda_{s_y}=0$) | $0.992 \pm 0.002$ | $3.511 \pm 0.449$ | $1.4\mathrm{e}{-2} \pm 3.0\mathrm{e}{-3}$ | $4.748 \pm 0.651$ | $1.272 \pm 0.198$ |
| (C) | ELBO+SIGReg ($\lambda_{s_x}=0, \lambda_{s_y}=10$) | $0.993 \pm 0.002$ | $1.999 \pm 0.058$ | $4.6\mathrm{e}{-3} \pm 4.0\mathrm{e}{-4}$ | $3.423 \pm 0.215$ | $0.189 \pm 0.030$ |
| (D) | ELBO+SIGReg | $0.992 \pm 0.002$ | $1.998 \pm 0.057$ | $4.6\mathrm{e}{-3} \pm 4.0\mathrm{e}{-4}$ | $3.423 \pm 0.216$ | $0.189 \pm 0.030$ |
| (E) | ELBO ($\alpha^{\mathrm{KL}}_{s_x}=0$) | $0.993 \pm 0.001$ | $4.051 \pm 0.517$ | $2.1\mathrm{e}{-2} \pm 3.4\mathrm{e}{-3}$ | $4.985 \pm 0.591$ | $1.589 \pm 0.201$ |
| (F) | ELBO ($\alpha^{\mathrm{KL}}_{s_y}=0$) | $0.983 \pm 0.003$ | $6.201 \pm 1.319$ | $2.8\mathrm{e}{-2} \pm 5.5\mathrm{e}{-3}$ | $6.288 \pm 1.636$ | $1.924 \pm 0.348$ |
| (G) | ELBO ($\alpha^{\mathrm{rec}}=\alpha^{\mathrm{gen}}=0$) | $0.543 \pm 0.033$ | $0.055 \pm 0.006$ | $4.2\mathrm{e}{-4} \pm 3.2\mathrm{e}{-5}$ | $0.428 \pm 0.021$ | $0.164 \pm 0.019$ |
| (H) | ELBO ($\alpha^{\mathrm{rec}}=\alpha^{\mathrm{gen}}=0$)+SIGReg | $0.821 \pm 0.067$ | $0.017 \pm 0.001$ | $1.6\mathrm{e}{-4} \pm 1.4\mathrm{e}{-5}$ | $0.239 \pm 0.018$ | $0.067 \pm 0.014$ |
| (I) | ELBO ($\alpha^{\mathrm{KL}}_{s_x}=\alpha^{\mathrm{KL}}_{z}=\alpha^{\mathrm{KL}}_{s_y}=0$) | $0.984 \pm 0.004$ | $20.276 \pm 11.331$ | $8.4\mathrm{e}{-2} \pm 4.3\mathrm{e}{-2}$ | $11.342 \pm 5.062$ | $4.871 \pm 2.150$ |
| (J) | ELBO ($\alpha^{\mathrm{KL}}_{s_x}=\alpha^{\mathrm{KL}}_{z}=\alpha^{\mathrm{KL}}_{s_y}=0$)+SIGReg | $0.983 \pm 0.001$ | $2.215 \pm 0.249$ | $3.4\mathrm{e}{-3} \pm 7.8\mathrm{e}{-4}$ | $2.825 \pm 0.271$ | $0.248 \pm 0.068$ |

*Figure 2.* Epoch-wise distribution diagnostics for selected experiments. We show how aggregated KL divergences, SIGReg-MSE, isotropy metrics, and conditional-prior coupling evolve during training.

**Datasets.** We evaluate learned representations on five real-world tabular datasets: Adult (AD), Covertype (CO), Electricity (EL), Credit Card (CC), and Bank Marketing (BM), as well as MNIST (treated as tabular features with controllable input corruption) and a fully-synthetic simulation dataset (SIM); more details are provided in Appendix C.1.1.

**Downstream and baseline predictors.** For each dataset, we compare strong raw-feature baselines to the same predictor architectures trained on embeddings produced by Var-T-JEPA and T-JEPA (details in Appendix D.5). Following Thimonier et al. (2025), we consider a range of strong and widely-used tabular predictors: MLP, DCNv2 (Wang et al., 2021), ResNet (He et al., 2016), AutoInt (Song et al., 2019), FT-Transformer (Gorishniy et al., 2021), and XGBoost (Chen & Guestrin, 2016).

**Selective evaluation via uncertainty.** Var-T-JEPA provides a per-sample uncertainty estimate; we report selective-

evaluation where the most uncertain $10\%$, $20\%$, or $50\%$ of samples are discarded before computing accuracy ("Var-T-JEPA (10%)", etc.), illustrating the coverage–accuracy trade-off when abstaining on low-confidence samples.

### 4.2.2. RESULTS

Table 2 summarizes downstream test accuracy across datasets and predictor families. Across the real-world tabular datasets (AD, CO, EL, CC, BM), Var-T-JEPA yields competitive embeddings for downstream classifiers, and selective evaluation exhibits a clear coverage–accuracy trade-off: discarding the most uncertain test samples improves performance on the retained subset across model families. In contrast, the deterministic T-JEPA baseline can suffer from representation collapse on some datasets–most notably CO and MNIST, where it degrades to near-random accuracy for several downstream models.

For the (semi-)synthetic MNIST and SIM datasets, Var-

*Table 2.* Downstream performance comparison across tabular datasets. Results (mean ± std over 5 downstream model runs; XGBoost single deterministic run) show test accuracy. AD=Adult, CO=Covertype, EL=Electricity, CC=Credit Card Default, BM=Bank Marketing. Light blue rows indicate selective evaluation (reduced coverage).

| Method | AD ↑ | CO ↑ | EL ↑ | CC ↑ | BM ↑ | MNIST ↑ | SIM ↑ |
|---|---|---|---|---|---|---|---|
| MLP | $0.849 \pm 1.4e^{-3}$ | $\mathbf{0.750} \pm 3.7e^{-3}$ | $0.781 \pm 3.6e^{-3}$ | $0.816 \pm 2.5e^{-3}$ | $0.898 \pm 1.4e^{-3}$ | $0.822 \pm 1.1e^{-2}$ | $\mathbf{0.823} \pm 7.5e^{-3}$ |
| +T-JEPA | $0.849 \pm 4.4e^{-3}$ | $0.408 \pm 2.3e^{-1}$ | $0.627 \pm 4.1e^{-2}$ | $0.774 \pm 0e^{-0}$ | $0.616 \pm 3.7e^{-1}$ | $0.113 \pm 6.2e^{-3}$ | $0.692 \pm 3.5e^{-2}$ |
| +Var-T-JEPA | $0.852 \pm 2.1e^{-3}$ | $0.679 \pm 2.9e^{-2}$ | $0.790 \pm 3.3e^{-3}$ | $0.818 \pm 1.5e^{-3}$ | $0.900 \pm 2.0e^{-3}$ | $0.822 \pm 1.8e^{-2}$ | $0.695 \pm 1.4e^{-2}$ |
| +Var-T-JEPA (10%) | $0.865 \pm 2.0e^{-3}$ | $0.696 \pm 2.8e^{-2}$ | $0.793 \pm 2.9e^{-3}$ | $0.827 \pm 1.4e^{-3}$ | $0.905 \pm 2.2e^{-3}$ | $0.838 \pm 1.6e^{-2}$ | $0.705 \pm 1.3e^{-2}$ |
| +Var-T-JEPA (20%) | $0.883 \pm 1.7e^{-3}$ | $0.697 \pm 2.2e^{-2}$ | $0.795 \pm 2.8e^{-3}$ | $0.835 \pm 1.2e^{-3}$ | $0.908 \pm 2.5e^{-3}$ | $0.856 \pm 1.9e^{-2}$ | $0.706 \pm 1.5e^{-2}$ |
| +Var-T-JEPA (50%) | $\mathbf{0.921} \pm 2.1e^{-3}$ | $0.706 \pm 1.4e^{-2}$ | $\mathbf{0.804} \pm 3.9e^{-3}$ | $0.841 \pm 7.0e^{-4}$ | $\mathbf{0.921} \pm 1.4e^{-3}$ | $\mathbf{0.901} \pm 2.1e^{-2}$ | $0.729 \pm 1.5e^{-2}$ |
| DCNv2 | $0.762 \pm 6.2e^{-4}$ | $0.757 \pm 6.2e^{-3}$ | $0.825 \pm 7.3e^{-4}$ | $0.814 \pm 1.1e^{-3}$ | $0.896 \pm 1.1e^{-3}$ | $0.875 \pm 4.5e^{-3}$ | $0.714 \pm 1.8e^{-3}$ |
| +T-JEPA | $0.851 \pm 1.5e^{-3}$ | $0.546 \pm 8.9e^{-3}$ | $0.769 \pm 2.3e^{-3}$ | $0.792 \pm 1.7e^{-2}$ | $0.895 \pm 3.1e^{-4}$ | $0.841 \pm 4.6e^{-2}$ | $\mathbf{0.772} \pm 3.7e^{-3}$ |
| +Var-T-JEPA | $0.851 \pm 1.8e^{-3}$ | $0.778 \pm 6.1e^{-3}$ | $0.830 \pm 1.9e^{-3}$ | $0.814 \pm 2.4e^{-3}$ | $0.896 \pm 6.9e^{-4}$ | $0.885 \pm 4.8e^{-3}$ | $0.720 \pm 7.8e^{-3}$ |
| +Var-T-JEPA (10%) | $0.864 \pm 1.7e^{-3}$ | $\mathbf{0.785} \pm 6.8e^{-3}$ | $0.831 \pm 2.6e^{-3}$ | $0.823 \pm 2.7e^{-3}$ | $0.903 \pm 7.9e^{-4}$ | $0.898 \pm 5.2e^{-3}$ | $0.728 \pm 7.2e^{-3}$ |
| +Var-T-JEPA (20%) | $0.882 \pm 1.6e^{-3}$ | $0.783 \pm 6.8e^{-3}$ | $0.831 \pm 2.1e^{-3}$ | $0.833 \pm 2.5e^{-3}$ | $0.907 \pm 1.2e^{-3}$ | $0.909 \pm 5.0e^{-3}$ | $0.726 \pm 7.5e^{-3}$ |
| +Var-T-JEPA (50%) | $\mathbf{0.920} \pm 1.3e^{-3}$ | $0.782 \pm 5.8e^{-3}$ | $\mathbf{0.831} \pm 2.7e^{-3}$ | $\mathbf{0.840} \pm 1.5e^{-3}$ | $\mathbf{0.918} \pm 1.6e^{-3}$ | $\mathbf{0.936} \pm 3.0e^{-3}$ | $0.757 \pm 5.4e^{-3}$ |
| ResNet | $0.849 \pm 1.9e^{-3}$ | $0.776 \pm 7.0e^{-3}$ | $0.823 \pm 2.5e^{-3}$ | $0.813 \pm 2.0e^{-3}$ | $0.897 \pm 2.3e^{-3}$ | $0.860 \pm 7.0e^{-3}$ | $\mathbf{0.878} \pm 5.9e^{-3}$ |
| +T-JEPA | $0.852 \pm 2.5e^{-3}$ | $0.540 \pm 6.3e^{-2}$ | $0.808 \pm 7.4e^{-3}$ | $0.817 \pm 3.7e^{-3}$ | $0.897 \pm 4.7e^{-3}$ | $0.854 \pm 3.0e^{-2}$ | $0.693 \pm 6.6e^{-2}$ |
| +Var-T-JEPA | $0.854 \pm 1.4e^{-3}$ | $0.779 \pm 1.1e^{-2}$ | $0.820 \pm 7.6e^{-3}$ | $0.813 \pm 5.0e^{-3}$ | $0.900 \pm 1.2e^{-3}$ | $0.871 \pm 1.3e^{-2}$ | $0.564 \pm 1.1e^{-1}$ |
| +Var-T-JEPA (10%) | $0.868 \pm 9.2e^{-4}$ | $0.785 \pm 1.1e^{-2}$ | $0.822 \pm 7.0e^{-3}$ | $0.824 \pm 2.9e^{-3}$ | $0.906 \pm 1.1e^{-3}$ | $0.887 \pm 9.7e^{-3}$ | $0.573 \pm 1.1e^{-1}$ |
| +Var-T-JEPA (20%) | $0.886 \pm 8.2e^{-4}$ | $0.782 \pm 1.3e^{-2}$ | $0.823 \pm 6.8e^{-3}$ | $0.833 \pm 2.5e^{-3}$ | $0.910 \pm 9.2e^{-4}$ | $0.900 \pm 8.7e^{-3}$ | $0.583 \pm 1.1e^{-1}$ |
| +Var-T-JEPA (50%) | $\mathbf{0.924} \pm 6.6e^{-4}$ | $\mathbf{0.787} \pm 1.4e^{-2}$ | $0.826 \pm 7.0e^{-3}$ | $0.841 \pm 2.2e^{-3}$ | $0.920 \pm 2.0e^{-3}$ | $\mathbf{0.936} \pm 5.2e^{-3}$ | $0.612 \pm 1.2e^{-1}$ |
| AutoInt | $0.761 \pm 7.8e^{-4}$ | $0.753 \pm 1.6e^{-2}$ | $0.754 \pm 1.1e^{-2}$ | $0.817 \pm 1.4e^{-2}$ | $0.901 \pm 1.5e^{-3}$ | $0.809 \pm 1.6e^{-2}$ | $\mathbf{0.897} \pm 1.9e^{-2}$ |
| +T-JEPA | $0.854 \pm 2.6e^{-3}$ | $0.448 \pm 1.3e^{-1}$ | $0.756 \pm 2.0e^{-2}$ | $0.804 \pm 3.2e^{-4}$ | $0.893 \pm 3.7e^{-3}$ | $0.817 \pm 7.4e^{-3}$ | $0.775 \pm 7.9e^{-3}$ |
| +Var-T-JEPA | $0.854 \pm 2.5e^{-3}$ | $0.752 \pm 4.5e^{-3}$ | $0.822 \pm 2.7e^{-3}$ | $0.816 \pm 1.3e^{-3}$ | $0.900 \pm 1.5e^{-3}$ | $0.810 \pm 4.9e^{-3}$ | $0.723 \pm 7.4e^{-3}$ |
| +Var-T-JEPA (10%) | $0.868 \pm 3.0e^{-3}$ | $\mathbf{0.756} \pm 4.2e^{-3}$ | $0.823 \pm 3.6e^{-3}$ | $0.824 \pm 1.5e^{-3}$ | $0.906 \pm 1.5e^{-3}$ | $0.825 \pm 2.0e^{-3}$ | $0.730 \pm 6.7e^{-3}$ |
| +Var-T-JEPA (20%) | $0.885 \pm 2.2e^{-3}$ | $0.754 \pm 4.7e^{-3}$ | $0.822 \pm 3.9e^{-3}$ | $0.834 \pm 1.1e^{-3}$ | $0.909 \pm 1.1e^{-3}$ | $0.841 \pm 1.9e^{-3}$ | $0.732 \pm 6.2e^{-3}$ |
| +Var-T-JEPA (50%) | $\mathbf{0.923} \pm 1.5e^{-3}$ | $0.749 \pm 6.2e^{-3}$ | $\mathbf{0.823} \pm 4.3e^{-3}$ | $\mathbf{0.840} \pm 9.6e^{-4}$ | $\mathbf{0.920} \pm 1.2e^{-3}$ | $\mathbf{0.880} \pm 4.2e^{-3}$ | $0.757 \pm 7.0e^{-3}$ |
| FT-Trans | $0.761 \pm 4.1e^{-4}$ | $0.747 \pm 9.1e^{-3}$ | $\mathbf{0.814} \pm 3.9e^{-3}$ | $0.820 \pm 1.0e^{-3}$ | $0.901 \pm 6.7e^{-4}$ | $0.877 \pm 2.4e^{-2}$ | $\mathbf{0.962} \pm 1.4e^{-3}$ |
| +T-JEPA | $0.854 \pm 1.0e^{-3}$ | $0.522 \pm 1.3e^{-2}$ | $0.705 \pm 4.7e^{-2}$ | $0.802 \pm 2.1e^{-3}$ | $0.886 \pm 3.7e^{-4}$ | $0.317 \pm 2.2e^{-1}$ | $0.723 \pm 1.2e^{-2}$ |
| +Var-T-JEPA | $0.852 \pm 1.2e^{-3}$ | $0.748 \pm 5.6e^{-3}$ | $0.798 \pm 4.2e^{-3}$ | $0.818 \pm 3.6e^{-4}$ | $0.900 \pm 4.1e^{-4}$ | $0.864 \pm 9.3e^{-3}$ | $0.716 \pm 7.7e^{-3}$ |
| +Var-T-JEPA (10%) | $0.867 \pm 1.4e^{-3}$ | $\mathbf{0.756} \pm 5.6e^{-3}$ | $0.800 \pm 3.7e^{-3}$ | $0.827 \pm 4.7e^{-4}$ | $0.905 \pm 6.3e^{-4}$ | $0.879 \pm 9.4e^{-3}$ | $0.726 \pm 8.2e^{-3}$ |
| +Var-T-JEPA (20%) | $0.885 \pm 1.4e^{-3}$ | $0.753 \pm 5.8e^{-3}$ | $0.802 \pm 3.5e^{-3}$ | $0.836 \pm 7.0e^{-4}$ | $0.909 \pm 6.6e^{-4}$ | $0.891 \pm 9.8e^{-3}$ | $0.727 \pm 6.5e^{-3}$ |
| +Var-T-JEPA (50%) | $\mathbf{0.923} \pm 1.2e^{-3}$ | $0.746 \pm 7.9e^{-3}$ | $0.809 \pm 2.1e^{-3}$ | $\mathbf{0.842} \pm 6.3e^{-4}$ | $\mathbf{0.922} \pm 8.8e^{-4}$ | $\mathbf{0.918} \pm 6.5e^{-3}$ | $0.748 \pm 2.5e^{-3}$ |
| XGBoost | 0.864 | 0.807 | **0.917** | 0.811 | 0.900 | 0.881 | 0.949 |
| +T-JEPA | 0.854 | 0.807 | 0.860 | 0.801 | 0.898 | 0.871 | 0.851 |
| +Var-T-JEPA | 0.855 | 0.809 | 0.888 | 0.806 | 0.904 | 0.872 | 0.945 |
| +Var-T-JEPA (10%) | 0.874 | **0.818** | 0.890 | 0.817 | 0.910 | 0.883 | 0.948 |
| +Var-T-JEPA (20%) | 0.893 | 0.818 | 0.891 | 0.823 | 0.912 | 0.889 | 0.951 |
| +Var-T-JEPA (50%) | **0.928** | 0.816 | 0.891 | **0.832** | **0.921** | **0.913** | **0.955** |

T-JEPA additionally produces uncertainty signals that are consistent with the underlying simulated corruption/ambiguity structure: Figure 3 visualizes (left) risk–coverage curves

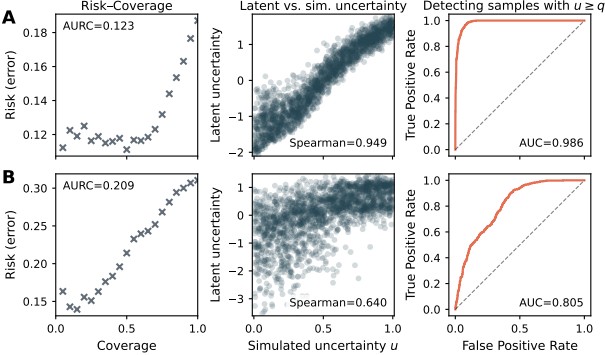

*Figure 3.* Uncertainty quantification on the MNIST (A) and SIM (B) datasets. Left: risk–coverage curve induced by abstaining on samples with highest latent uncertainty. Middle: standardized latent uncertainty versus simulated uncertainty. Right: ROC curve for detecting high-ambiguity samples from latent uncertainty.

induced by abstaining on samples with highest latent uncertainty, (middle) a positive association between standardized latent uncertainty and the simulated uncertainty score, and (right) ROC curves showing that latent uncertainty can identify high-ambiguity samples (defined by a high-quantile threshold on the simulated uncertainty score). These diagnostics complement the selective-evaluation results in Table 2 by showing that the learned latent uncertainty is not only useful for abstention, but also aligned with the known uncertainty signal in these controlled settings. Additional experimental results, including sensitivity analyses, are provided in Appendix B.

## 5. Discussion

We provide a novel reinterpretation of the JEPA design pattern as variational inference in a coupled latent-variable model: the predictor is a learned conditional prior $p_\theta(s_y \mid s_x, z)$, and ad hoc costs are indirect forms of distributional control that the ELBO regularizes. This clarifies that the common "JEPA vs. generative modeling" dichotomy is

largely a matter of framing: JEPA occupies a particular (implicit, often deterministic) point in the design space of latent-variable generative models. By making this latent generative structure explicit, we unify predictive and generative self-supervised learning within a single principled framework. Empirically, training with the complete ELBO yields usable representations without heuristic anti-collapse objectives (e.g., auxiliary regularizers such as EMA or distribution-matching penalties) and enables rigorous uncertainty estimates from posterior covariances. We introduce Var-T-JEPA as a concrete implementation for tabular data, and in downstream tabular evaluation it produces competitive embeddings across various datasets and supports selective prediction based on estimated latent uncertainty. More broadly, we observe that per-sample KL terms to fixed priors drive aggregated distributional behavior comparable to explicit distributional regularizers such as SIGReg, while leaving the target latent governed by its learned conditional prior. Future work includes scaling to vision and video, and extending to settings where target observations are absent at test time, making conditional generation central.

## Acknowledgements

MG is supported by the EPSRC Centre for Doctoral Training in Health Data Science (EP/S02428X/1). CY is supported by a UKRI Turing AI Acceleration Fellowship (EP/V023233/2).

## Impact Statement

This paper presents work whose goal is to advance the field of ML. There are many potential societal consequences of our work, none which we feel must be specifically highlighted here.

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

# A. Var-T-JEPA: Variational Joint Embedding Predictive Architecture for Tabular Data

## A.1. Overview: Neural Architecture and Modeling Strategies

The following provides a full conceptual overview of Var-T-JEPA. Detailed implementation specifications are deferred to Appendix D.2.

Tabular data differs fundamentally from image or text modalities often encountered in JEPA applications in several aspects: (1) features are inherently heterogeneous, mixing numerical and categorical variables with different scales and distributions; (2) the notion of spatial or temporal locality is absent, requiring alternative masking strategies; (3) feature interactions are often complex and non-obvious, making the predictive task particularly suitable for representation learning approaches like JEPA.

Our tabular Var-T-JEPA pipeline broadly builds on modeling strategies established by Thimonier et al. (2025), who introduced T-JEPA as a standard (non-variational) JEPA implementation for tabular data. The model architecture of our tabular Var-T-JEPA model is depicted in Fig. 4, which translates the theoretical framework from Section 3 into a practical neural architecture tailored for tabular data.

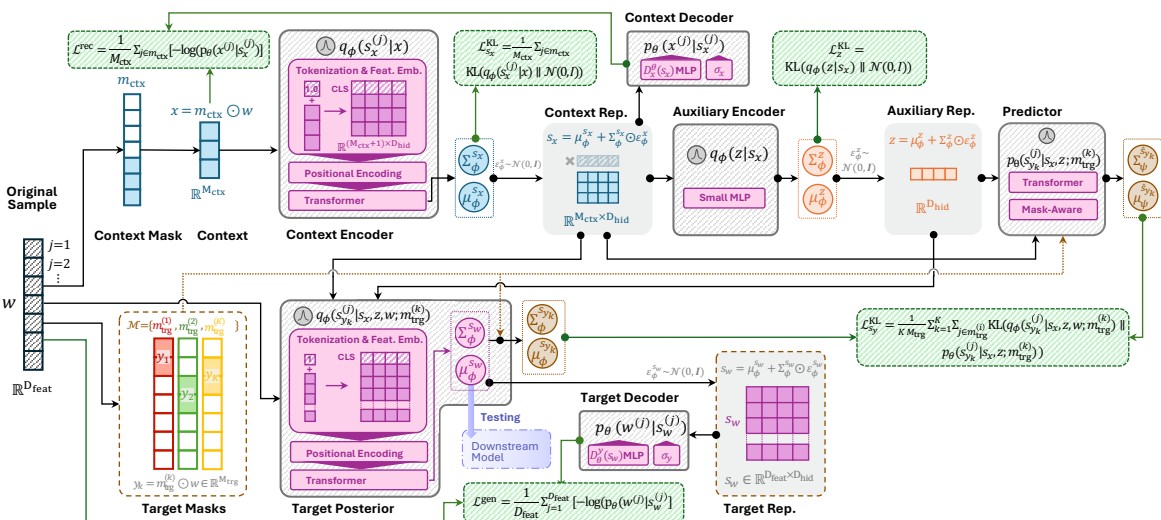

*Figure 4.* Architectural overview of Var-T-JEPA for tabular data. The model implements the theoretical framework through specialized components: (1) the context encoder processes masked tabular features to produce variational context representations $q_\phi(s_x^{(j)}|x)$; (2) the auxiliary encoder infers predictive latents $q_\phi(z|s_x)$; (3) the target posterior prepends context and auxiliary latents as special tokens before feature processing $q_\phi(s_{y_k}^{(j)}|s_x, z, w; m_{\text{trg}}^{(k)})$; (4) the predictor generates target representations $p_\theta(s_{y_k}^{(j)}|s_x, z; m_{\text{trg}}^{(k)})$, and (5) the decoders reconstruct original tabular features from latent representations.

## A.2. Tabular Feature Masking Strategy

Following the T-JEPA masking framework, we partition the complete feature vector $w \in \mathbb{R}^{D_{\text{feat}}}$ into context and target subsets at the feature level; in this formulation $w$ is the full (unmasked) target view, while $y_k$ are masked target sub-views and $x$ the context view derived from $w$. Given $D_{\text{feat}}$ features, we create context masks $m_{\text{ctx}} \in \{0,1\}^{D_{\text{feat}}}$ and target masks $m_{\text{trg}}^{(i)} \in \{0,1\}^{D_{\text{feat}}}$ where:

$$x = m_{\text{ctx}} \odot w, \quad \|m_{\text{ctx}}\|_0 = M_{\text{ctx}} \tag{12}$$

$$y_k = m_{\text{trg}}^{(k)} \odot w, \quad \|m_{\text{trg}}^{(k)}\|_0 = M_{\text{trg}}, \quad k = 1, \dots, K \tag{13}$$

The mask sizes are sampled uniformly for each batch-optimization step: $M_{\text{ctx}} \sim \text{Uniform}[\lfloor D_{\text{feat}} \cdot r_{\text{ctx}}^{\min} \rfloor, \lfloor D_{\text{feat}} \cdot r_{\text{ctx}}^{\max} \rfloor]$ and $M_{\text{trg}} \sim \text{Uniform}[\lfloor D_{\text{feat}} \cdot r_{\text{trg}}^{\min} \rfloor, \lfloor D_{\text{feat}} \cdot r_{\text{trg}}^{\max} \rfloor]$, where the ratio parameters define the minimum and maximum shares of features to mask. During training, we typically generate one context view and $K$ target predictions per sample, ensuring that $M_{\text{ctx}} + M_{\text{trg}} \leq D_{\text{feat}}$ to maintain non-overlapping masks.

### A.3. Variational Network Components

Our tabular Var-T-JEPA architecture consists of six key variational network components that implement the theoretical framework for heterogeneous tabular data. Each feature is indexed by $j \in \{1, 2, \ldots, D_{\text{feat}}\}$ where $D_{\text{feat}}$ denotes the total number of features in the dataset. Features are categorized as either numerical ($j \in \mathcal{I}_{\text{num}}$) or categorical ($j \in \mathcal{I}_{\text{cat}}$). More detailed implementation specifications are provided in Appendix D.2.

**(I) Context Encoder.** Following Thimonier et al. (2025), $q_\phi(s_x^{(j)}|x)$ processes masked tabular features through feature tokenization, transformer encoding, and variational output heads. Each feature is embedded with type and positional information, then processed by a transformer with prepended CLS tokens, yielding contextualized representations that parameterize the context latent distribution

$$h_{\text{ctx}} = f_\phi^{\text{ctx}}(x), \tag{14}$$

$$q_\phi(s_x^{(j)}|x) = \mathcal{N}(s_x^{(j)}; \mu_\phi^{s_x}(h_{\text{ctx}})^{(j)}, \Sigma_\phi^{s_x}(h_{\text{ctx}})^{(j)}). \tag{15}$$

Here, $f_\phi^{\text{ctx}}(x)$ is the transformer-based encoder and $\mu_\phi^{s_x}$ and $\Sigma_\phi^{s_x}$ are linear projections applied to the encoder output.

**(II) Auxiliary Encoder.** $q_\phi(z|s_x)$ infers auxiliary predictive latents from (pooled) context representations $s_x$ through a compact MLP, maintaining the JEPA principle of preventing target information leakage:

$$q_\phi(z|s_x) = \mathcal{N}(z; \mu_\phi^z(s_x), \Sigma_\phi^z(s_x)) \tag{16}$$

**(III) Target Posterior.** $q_\phi(s_{y_k}^{(j)}|s_x, z, w; m_{\text{trg}}^{(k)})$ conditions on context latents, auxiliary latents, and complete raw features using a similar encoder design as the context encoder, but extended to incorporate latent tokens to condition on $s_x$ and $z$.

$$h_{\text{trg}} = f_\phi^{\text{trg}}(s_x, z, w), \tag{17}$$

$$q_\phi(s_{y_k}^{(j)}|s_x, z, w; m_{\text{trg}}^{(k)}) = \mathcal{N}(s_w^{(j)}; \mu_\phi^{s_w}(h_{\text{trg}})^{(j)}, \Sigma_\phi^{s_w}(h_{\text{trg}})^{(j)}) \text{ for } j \in m_{\text{trg}}^{(k)}. \tag{18}$$

Here, $f_\phi^{\text{trg}}(s_x, z, w)$ is the transformer-based encoder, and $\mu_\phi^{s_w}$ and $\Sigma_\phi^{s_w}$ are linear projections applied to the encoder output. $s_{y_k}^{(j)} = s_w^{(j)} : j \in m_{\text{trg}}^{(k)}$ are the masked target representations.

**(IV) Predictive Model.** $p_\theta(s_{y_k}^{(j)}|s_x, z; m_{\text{trg}}^{(k)})$ generates target latent distributions without access to target observations:

$$h_{\text{pred}} = g_\theta(s_x, z, m_{\text{trg}}^{(k)}), \tag{19}$$

$$p_\theta(s_{y_k}^{(j)}|s_x, z; m_{\text{trg}}^{(k)}) = \mathcal{N}(s_{y_k}^{(j)}; \mu_\theta^{s_y}(h_{\text{pred}})^{(j)}, \Sigma_\theta^{s_y}(h_{\text{pred}})^{(j)}). \tag{20}$$

Here, $g_\theta(s_x, z, m_{\text{trg}}^{(k)})$ involves another transformer architecture which is conditioned on the target mask $m_{\text{trg}}^{(k)}$ through positional embeddings and learnable mask tokens. Its output is used to generate distributional parameters for the target latent $s_y$ via linear projections.

**(V) Context and (VI) Target Decoders.** The decoders reconstruct masked features using feature-specific MLPs, $u_\theta^{(j)}$. Let $\pi \in \{x, w\}$ denote any observation (context or full target) with corresponding latent representation $s_\pi$. Each decoder reconstructs a masked feature independently according to the unified formulation:

$$p_\theta(\pi^{(j)}|s_\pi^{(j)}) = \begin{cases} \mathcal{N}(\pi^{(j)}; u_\theta^{(j)}(s_\pi^{(j)}), \sigma_\pi^2) & \text{if } j \in \mathcal{I}_{\text{num}} \\ \text{Categorical}(\pi^{(j)}; \text{softmax}(u_\theta^{(j)}(s_\pi^{(j)}))) & \text{if } j \in \mathcal{I}_{\text{cat}} \end{cases} \tag{21}$$

The full reconstruction distribution factorizes as $p_\theta(\pi|s_\pi) = \prod_{j \in m_\pi} p_\theta(\pi^{(j)}|s_\pi^{(j)})$, where $m_\pi$ denotes the feature index set for observation $\pi$ (i.e., $m_{\text{ctx}}$ for context $x$, and all feature indices for $w$ under full-target reconstruction). For numerical features, the reconstruction noise $\sigma_\pi^2$ is one of two globally learned parameters: $\sigma_x^2$ for the context path ($\pi = x$), and a shared $\sigma_y^2$ for the target path ($\pi = w$).

A.3.1. LOSS COMPUTATION

Adapting Var-JEPA's general loss function (10) to feature-level masking strategies introduced by Thimonier et al. (2025), we obtain the following sample-level loss terms for Var-T-JEPA:

$$\mathcal{L}^{\text{rec}} = \frac{1}{M_{\text{ctx}}} \sum_{j \in m_{\text{ctx}}} -\log p_\theta(x^{(j)}|s_x^{(j)}) \tag{22}$$

$$\mathcal{L}^{\text{gen}} = \frac{1}{D_{\text{feat}}} \sum_{j=1}^{D_{\text{feat}}} -\log p_\theta(w^{(j)} \mid s_w^{(j)}) \tag{23}$$

$$\mathcal{L}^{\text{KL}}_{s_x} = \frac{1}{M_{\text{ctx}}} \sum_{j \in m_{\text{ctx}}} \text{KL}\big(q_\phi(s_x^{(j)}|x)\|\mathcal{N}(0, I)\big) \tag{24}$$

$$\mathcal{L}^{\text{KL}}_z = \text{KL}\big(q_\phi(z|s_x)\|\mathcal{N}(0, I)\big) \tag{25}$$

$$\mathcal{L}^{\text{KL}}_{s_y} = \frac{1}{K \cdot M_{\text{trg}}} \sum_{k=1}^{K} \sum_{j \in m_{\text{trg}}^{(k)}} \text{KL}\big(q_\phi(s_{y_k}^{(j)}|s_x, z, w; m_{\text{trg}}^{(k)})\|p_\theta(s_{y_k}^{(j)}|s_x, z; m_{\text{trg}}^{(k)})\big) \tag{26}$$

## A.4. Outlook: Extension to Vision and Video

We implemented Var-JEPA as Var-T-JEPA to provide a compute-efficient proof-of-concept in a setting where careful ablations and uncertainty analyses are tractable. However, the formulation is not specific to tabular data and can be transferred directly to visual modalities in close analogy to I-JEPA (Assran et al., 2023). Concretely, one can (i) tokenize images/videos into patch or tubelet sequences, (ii) define a masked context view $x$ and masked target view $y$, (iii) parameterize $q_\phi(s_x \mid x)$ and $q_\phi(s_y \mid s_x, z, y)$ with ViT-based encoders and $p_\theta(s_y \mid s_x, z)$ with a lightweight predictor (as in I-JEPA), and (iv) add a decoder $p_\theta(y \mid s_y)$ (e.g., Gaussian pixels or discrete token likelihoods) to obtain the ELBO objective. This yields a generative, uncertainty-aware analogue of JEPA-style representation prediction for vision/video, while retaining the same conditional-prior interpretation of the predictor.

# B. Additional Experimental Results

Our experimental evaluation intentionally focuses on validating our conceptual contribution (bridging JEPA and variational inference) within the JEPA framework–emphasizing collapse prevention, uncertainty quantification, and the relationship between per-sample and aggregated regularization–rather than comprehensive benchmarking against the broader self-supervised learning landscape. To complement the accuracy results reported in the main text, we provide F1-score evaluations, sensitivity analyses, and a comparison of uncertainty-quantification methods below.

## B.1. Additional Downstream Metrics: F1 Score

Table 3 reports downstream *macro* F1-scores (mean ± std over 5 downstream model runs) for the same set of tabular datasets and predictor families as Table 2. As in the main table, we report results on raw features, on T-JEPA embeddings, and on Var-T-JEPA embeddings (including selective evaluation where the most uncertain fraction of test samples is discarded).

## B.2. Sensitivity Analysis

B.2.1. SENSITIVITY ANALYSIS ON $\alpha$

We perform a sensitivity study on Adult by varying the end weights of the KL terms for $s_x$ and $s_y$ (i.e., $\alpha^{\text{KL}}_{s_x}$ and $\alpha^{\text{KL}}_{s_y}$) and the reconstruction/generation weights (i.e., $\alpha^{\text{rec}}$ and $\alpha^{\text{gen}}$) around the baseline setting. Table 4 shows that downstream performance is broadly robust across these ablations (i.e., not strongly sensitive to precise tuning), and that selective evaluation consistently improves accuracy across all settings as more uncertain samples are filtered out.

B.2.2. SENSITIVITY ANALYSIS ON OTHER MODEL PARAMETERS

We additionally study sensitivity to model capacity and optimization hyperparameters by varying the transformer hidden dimension $d$ (token/embedding width), number of layers $L$, predictor feedforward dimension $\text{ff}_p$ (i.e., the MLP width in the predictor block), learning rate $\eta$, and batch size $B$. Table 5 shows that downstream accuracy remains broadly stable for most settings, while some changes (e.g., substantially deeper encoders or smaller batch size) can reduce performance. As in the $\alpha$ sensitivity study, selective evaluation improves accuracy consistently across all ablations as more uncertain samples are filtered out.

*Table 3.* Downstream performance comparison across tabular datasets using macro F1-score. Results (mean $\pm$ std over 5 downstream model runs; XGBoost single deterministic run) show test macro F1-score. AD=Adult, CO=Covertype, EL=Electricity, CC=Credit Card, BM=Bank Marketing. Light blue rows indicate selective evaluation (reduced coverage).

| Method | AD ↑ | CO ↑ | EL ↑ | CC ↑ | BM ↑ | MNIST ↑ | SIM ↑ |
|---|---|---|---|---|---|---|---|
| MLP | $0.845\pm3.7e^{-3}$ | $\mathbf{0.747}\pm5.0e^{-3}$ | $0.780\pm3.5e^{-3}$ | $0.792\pm6.9e^{-3}$ | $0.887\pm4.8e^{-3}$ | $0.822\pm1.1e^{-2}$ | $\mathbf{0.821}\pm9.0e^{-3}$ |
| +T-JEPA | $0.842\pm9.0e^{-3}$ | $0.274\pm1.5e^{-1}$ | $0.545\pm7.4e^{-2}$ | $0.675\pm0e^{-0}$ | $0.583\pm3.9e^{-1}$ | $0.023\pm2.3e^{-3}$ | $0.662\pm3.3e^{-2}$ |
| +Var-T-JEPA | $0.845\pm3.3e^{-3}$ | $0.657\pm3.8e^{-2}$ | $0.789\pm2.2e^{-3}$ | $\mathbf{0.797}\pm4.1e^{-3}$ | $0.889\pm3.1e^{-3}$ | $0.823\pm1.7e^{-2}$ | $0.659\pm3.3e^{-2}$ |
| +Var-T-JEPA (10%) | $0.859\pm3.0e^{-3}$ | $0.678\pm3.4e^{-2}$ | $0.792\pm2.0e^{-3}$ | $0.792\pm5.6e^{-3}$ | $0.893\pm3.5e^{-3}$ | $0.840\pm1.5e^{-2}$ | $0.671\pm3.1e^{-2}$ |
| +Var-T-JEPA (20%) | $0.877\pm2.4e^{-3}$ | $0.680\pm3.0e^{-2}$ | $0.794\pm2.3e^{-3}$ | $0.792\pm4.3e^{-3}$ | $0.895\pm4.0e^{-3}$ | $0.857\pm1.8e^{-2}$ | $0.671\pm3.2e^{-2}$ |
| +Var-T-JEPA (50%) | $\mathbf{0.915}\pm2.6e^{-3}$ | $0.692\pm1.7e^{-2}$ | $\mathbf{0.803}\pm3.4e^{-3}$ | $0.780\pm1.9e^{-3}$ | $\mathbf{0.907}\pm3.9e^{-3}$ | $\mathbf{0.901}\pm2.0e^{-2}$ | $0.691\pm3.7e^{-2}$ |
| DCNv2 | $0.684\pm4.5e^{-3}$ | $0.749\pm6.2e^{-3}$ | $0.824\pm7.3e^{-4}$ | $0.794\pm3.0e^{-3}$ | $0.890\pm2.1e^{-3}$ | $0.875\pm4.3e^{-3}$ | $0.673\pm5.0e^{-3}$ |
| +T-JEPA | $0.846\pm1.7e^{-3}$ | $0.443\pm5.3e^{-2}$ | $0.764\pm1.9e^{-3}$ | $0.737\pm5.7e^{-2}$ | $0.878\pm1.4e^{-3}$ | $0.840\pm4.7e^{-2}$ | $\mathbf{0.759}\pm4.2e^{-3}$ |
| +Var-T-JEPA | $0.848\pm1.5e^{-3}$ | $0.775\pm6.9e^{-3}$ | $0.830\pm1.9e^{-3}$ | $\mathbf{0.794}\pm1.3e^{-3}$ | $0.891\pm5.9e^{-4}$ | $0.884\pm4.8e^{-3}$ | $0.703\pm8.8e^{-3}$ |
| +Var-T-JEPA (10%) | $0.861\pm1.2e^{-3}$ | $\mathbf{0.783}\pm7.0e^{-3}$ | $0.831\pm2.6e^{-3}$ | $0.790\pm1.4e^{-3}$ | $0.896\pm6.6e^{-4}$ | $0.897\pm5.2e^{-3}$ | $0.711\pm8.3e^{-3}$ |
| +Var-T-JEPA (20%) | $0.878\pm1.3e^{-3}$ | $0.780\pm7.1e^{-3}$ | $0.831\pm2.1e^{-3}$ | $0.791\pm1.9e^{-3}$ | $0.900\pm1.1e^{-3}$ | $0.908\pm5.0e^{-3}$ | $0.707\pm8.6e^{-3}$ |
| +Var-T-JEPA (50%) | $\mathbf{0.916}\pm7.9e^{-4}$ | $0.778\pm5.7e^{-3}$ | $\mathbf{0.831}\pm2.6e^{-3}$ | $0.780\pm1.6e^{-3}$ | $\mathbf{0.909}\pm1.8e^{-3}$ | $\mathbf{0.936}\pm3.0e^{-3}$ | $0.735\pm6.1e^{-3}$ |
| ResNet | $0.844\pm1.8e^{-3}$ | $0.776\pm7.3e^{-2}$ | $0.823\pm2.6e^{-3}$ | $\mathbf{0.794}\pm1.5e^{-3}$ | $0.890\pm4.0e^{-3}$ | $0.860\pm7.1e^{-3}$ | $\mathbf{0.877}\pm5.9e^{-3}$ |
| +T-JEPA | $0.845\pm4.8e^{-3}$ | $0.499\pm8.7e^{-2}$ | $0.807\pm8.8e^{-3}$ | $0.794\pm7.9e^{-3}$ | $0.893\pm3.8e^{-3}$ | $0.856\pm2.9e^{-2}$ | $0.689\pm5.8e^{-2}$ |
| +Var-T-JEPA | $0.849\pm2.4e^{-3}$ | $0.778\pm1.2e^{-2}$ | $0.820\pm7.2e^{-3}$ | $0.791\pm1.0e^{-2}$ | $0.894\pm2.6e^{-3}$ | $0.872\pm1.2e^{-2}$ | $0.549\pm1.2e^{-1}$ |
| +Var-T-JEPA (10%) | $0.864\pm1.6e^{-3}$ | $0.784\pm1.2e^{-2}$ | $0.822\pm6.8e^{-3}$ | $0.789\pm7.0e^{-3}$ | $0.899\pm2.0e^{-3}$ | $0.888\pm9.2e^{-3}$ | $0.560\pm1.1e^{-1}$ |
| +Var-T-JEPA (20%) | $0.881\pm1.2e^{-3}$ | $0.781\pm1.3e^{-2}$ | $0.823\pm6.7e^{-3}$ | $0.789\pm5.1e^{-3}$ | $0.903\pm1.9e^{-3}$ | $0.900\pm8.5e^{-3}$ | $0.570\pm1.1e^{-1}$ |
| +Var-T-JEPA (50%) | $\mathbf{0.920}\pm8.5e^{-4}$ | $\mathbf{0.784}\pm1.3e^{-2}$ | $\mathbf{0.826}\pm7.5e^{-3}$ | $0.780\pm2.1e^{-3}$ | $\mathbf{0.912}\pm1.0e^{-3}$ | $\mathbf{0.936}\pm5.3e^{-3}$ | $0.602\pm1.1e^{-1}$ |
| AutoInt | $0.687\pm6.3e^{-3}$ | $0.752\pm1.7e^{-2}$ | $0.755\pm1.1e^{-2}$ | $\mathbf{0.798}\pm1.2e^{-3}$ | $0.896\pm2.5e^{-3}$ | $0.809\pm1.6e^{-2}$ | $\mathbf{0.897}\pm1.9e^{-2}$ |
| +T-JEPA | $0.849\pm2.9e^{-3}$ | $0.399\pm1.6e^{-1}$ | $0.745\pm3.1e^{-2}$ | $0.777\pm7.8e^{-4}$ | $0.871\pm1.2e^{-2}$ | $0.816\pm7.9e^{-3}$ | $0.768\pm8.4e^{-3}$ |
| +Var-T-JEPA | $0.848\pm3.1e^{-3}$ | $0.750\pm4.4e^{-3}$ | $0.821\pm2.8e^{-3}$ | $0.796\pm2.9e^{-3}$ | $0.894\pm1.3e^{-3}$ | $0.810\pm5.5e^{-3}$ | $0.711\pm9.3e^{-3}$ |
| +Var-T-JEPA (10%) | $0.863\pm3.6e^{-3}$ | $\mathbf{0.755}\pm4.2e^{-3}$ | $0.822\pm3.8e^{-3}$ | $0.791\pm3.1e^{-3}$ | $0.898\pm1.4e^{-3}$ | $0.824\pm2.0e^{-3}$ | $0.719\pm8.3e^{-3}$ |
| +Var-T-JEPA (20%) | $0.880\pm2.8e^{-3}$ | $0.752\pm4.3e^{-3}$ | $0.822\pm4.1e^{-3}$ | $0.793\pm2.2e^{-3}$ | $0.902\pm1.4e^{-3}$ | $0.840\pm1.7e^{-3}$ | $0.720\pm7.9e^{-3}$ |
| +Var-T-JEPA (50%) | $\mathbf{0.918}\pm2.2e^{-3}$ | $0.745\pm6.3e^{-3}$ | $\mathbf{0.823}\pm4.5e^{-3}$ | $0.781\pm1.6e^{-3}$ | $\mathbf{0.912}\pm1.1e^{-3}$ | $\mathbf{0.879}\pm3.8e^{-3}$ | $0.741\pm1.0e^{-2}$ |
| FT-Trans | $0.687\pm4.9e^{-3}$ | $0.741\pm1.0e^{-2}$ | $\mathbf{0.813}\pm4.0e^{-3}$ | $\mathbf{0.799}\pm1.0e^{-3}$ | $0.894\pm1.3e^{-3}$ | $0.877\pm2.3e^{-2}$ | $\mathbf{0.962}\pm1.4e^{-3}$ |
| +T-JEPA | $0.847\pm1.1e^{-3}$ | $0.417\pm8.5e^{-2}$ | $0.700\pm4.7e^{-2}$ | $0.777\pm2.0e^{-3}$ | $0.851\pm1.0e^{-3}$ | $0.269\pm2.7e^{-1}$ | $0.696\pm2.0e^{-2}$ |
| +Var-T-JEPA | $0.847\pm2.2e^{-3}$ | $0.743\pm6.5e^{-3}$ | $0.797\pm4.6e^{-3}$ | $0.798\pm1.3e^{-3}$ | $0.893\pm9.2e^{-4}$ | $0.864\pm9.4e^{-3}$ | $0.701\pm8.0e^{-3}$ |
| +Var-T-JEPA (10%) | $0.862\pm2.3e^{-3}$ | $\mathbf{0.752}\pm5.6e^{-3}$ | $0.799\pm4.0e^{-3}$ | $0.793\pm9.1e^{-4}$ | $0.897\pm9.1e^{-4}$ | $0.879\pm9.6e^{-3}$ | $0.712\pm8.5e^{-3}$ |
| +Var-T-JEPA (20%) | $0.879\pm2.0e^{-3}$ | $0.750\pm5.7e^{-3}$ | $0.801\pm3.8e^{-3}$ | $0.794\pm1.6e^{-3}$ | $0.900\pm1.3e^{-3}$ | $0.891\pm9.9e^{-3}$ | $0.712\pm6.8e^{-3}$ |
| +Var-T-JEPA (50%) | $\mathbf{0.917}\pm2.0e^{-3}$ | $0.743\pm7.9e^{-3}$ | $0.809\pm2.2e^{-3}$ | $0.782\pm1.9e^{-3}$ | $\mathbf{0.912}\pm1.3e^{-3}$ | $\mathbf{0.918}\pm6.7e^{-3}$ | $0.732\pm3.6e^{-3}$ |
| XGBoost | 0.860 | 0.805 | **0.917** | **0.791** | 0.894 | 0.881 | 0.948 |
| +T-JEPA | 0.850 | 0.804 | 0.860 | 0.783 | 0.889 | 0.870 | 0.849 |
| +Var-T-JEPA | 0.850 | 0.807 | 0.888 | 0.787 | 0.897 | 0.871 | 0.945 |
| +Var-T-JEPA (10%) | 0.870 | **0.816** | 0.890 | 0.788 | 0.901 | 0.883 | 0.948 |
| +Var-T-JEPA (20%) | 0.888 | 0.815 | 0.891 | 0.788 | 0.903 | 0.889 | 0.951 |
| +Var-T-JEPA (50%) | **0.923** | 0.813 | 0.891 | 0.784 | **0.911** | **0.913** | **0.955** |

*Table 4.* Adult sensitivity analysis for Var-T-JEPA end-weight settings. Columns correspond to ablations varying $(\alpha_{s_x}^{\mathrm{KL}}, \alpha_{s_y}^{\mathrm{KL}}, \alpha^{\mathrm{rec}}, \alpha^{\mathrm{gen}})$; we report downstream test accuracy (mean $\pm$ std over 5 downstream runs) for MLP and ResNet probes, including selective evaluation.

| | Base | Low KL | High KL | Low recon. | High recon. | High KL + low recon. | Low KL + high recon. |
|---|---|---|---|---|---|---|---|
| $\alpha_{s_x}^{\mathrm{KL}}$ | $10^{-4}$ | $10^{-5}$ | $10^{-3}$ | $10^{-4}$ | $10^{-4}$ | $10^{-3}$ | $10^{-5}$ |
| $\alpha_{s_y}^{\mathrm{KL}}$ | $10^{-5}$ | $10^{-6}$ | $10^{-4}$ | $10^{-5}$ | $10^{-5}$ | $10^{-4}$ | $10^{-6}$ |
| $\alpha^{\mathrm{rec}}$ | $10^{-1}$ | $10^{-1}$ | $10^{-1}$ | $10^{-2}$ | $1$ | $10^{-2}$ | $1$ |
| $\alpha^{\mathrm{gen}}$ | $1$ | $1$ | $1$ | $10^{-1}$ | $10$ | $10^{-1}$ | $10$ |
| MLP + Var-T-JEPA | $0.852\pm2.1e^{-3}$ | $0.850\pm1.2e^{-3}$ | $0.826\pm2.7e^{-3}$ | $0.834\pm2.7e^{-3}$ | $0.852\pm2.3e^{-3}$ | $0.849\pm1.1e^{-3}$ | $0.851\pm1.2e^{-3}$ |
| MLP + Var-T-JEPA (10%) | $0.865\pm2.0e^{-3}$ | $0.864\pm9.6e^{-4}$ | $0.839\pm2.6e^{-3}$ | $0.850\pm2.0e^{-3}$ | $0.868\pm1.7e^{-3}$ | $0.863\pm1.3e^{-3}$ | $0.860\pm1.1e^{-3}$ |
| MLP + Var-T-JEPA (20%) | $0.883\pm1.7e^{-3}$ | $0.880\pm1.4e^{-3}$ | $0.852\pm2.4e^{-3}$ | $0.868\pm2.2e^{-3}$ | $0.879\pm1.9e^{-3}$ | $0.873\pm1.2e^{-3}$ | $0.878\pm8.5e^{-4}$ |
| MLP + Var-T-JEPA (50%) | $0.921\pm2.1e^{-3}$ | $0.920\pm8.7e^{-4}$ | $0.890\pm2.8e^{-3}$ | $0.888\pm3.0e^{-3}$ | $0.906\pm2.5e^{-3}$ | $0.888\pm1.4e^{-3}$ | $0.928\pm7.9e^{-4}$ |
| ResNet + Var-T-JEPA | $0.854\pm1.4e^{-3}$ | $0.852\pm1.8e^{-3}$ | $0.827\pm8.5e^{-4}$ | $0.836\pm3.1e^{-3}$ | $0.852\pm1.1e^{-3}$ | $0.853\pm1.6e^{-3}$ | $0.853\pm1.4e^{-3}$ |
| ResNet + Var-T-JEPA (10%) | $0.868\pm9.2e^{-4}$ | $0.867\pm1.4e^{-3}$ | $0.839\pm1.3e^{-3}$ | $0.851\pm3.1e^{-3}$ | $0.868\pm1.1e^{-3}$ | $0.867\pm1.8e^{-3}$ | $0.863\pm1.6e^{-3}$ |
| ResNet + Var-T-JEPA (20%) | $0.886\pm8.2e^{-4}$ | $0.883\pm1.9e^{-3}$ | $0.852\pm1.5e^{-3}$ | $0.869\pm3.2e^{-3}$ | $0.878\pm1.3e^{-3}$ | $0.876\pm1.1e^{-3}$ | $0.881\pm1.1e^{-3}$ |
| ResNet + Var-T-JEPA (50%) | $0.924\pm6.6e^{-4}$ | $0.921\pm8.6e^{-4}$ | $0.890\pm2.5e^{-3}$ | $0.888\pm4.3e^{-3}$ | $0.906\pm1.3e^{-3}$ | $0.891\pm1.4e^{-3}$ | $0.929\pm1.4e^{-3}$ |

*Table 5.* Adult sensitivity analysis for Var-T-JEPA architecture and optimization settings. Columns correspond to model/optimizer ablations varying $(d, L, \mathrm{ff}_p, \eta, B)$. We report downstream test accuracy (mean $\pm$ std over 5 downstream runs) for MLP and ResNet probes, including selective evaluation (light blue rows).

| | Shallow | Low pred.-FF | Narrow | High pred.-FF | Deep | Low LR | Batch size 256 |
|---|---|---|---|---|---|---|---|
| $d$ (hidden dim.) | 64 | 64 | 32 | 64 | 64 | 64 | 64 |
| $L$ (layers) | 4 | 8 | 8 | 8 | 16 | 8 | 8 |
| $\mathrm{ff}_p$ (pred.) | 256 | 128 | 128 | 512 | 256 | 256 | 256 |
| $\eta$ (LR) | $10^{-3}$ | $10^{-3}$ | $10^{-3}$ | $10^{-3}$ | $10^{-3}$ | $5 \times 10^{-4}$ | $10^{-3}$ |
| $B$ (batch) | 512 | 512 | 512 | 512 | 512 | 512 | 256 |
| MLP + Var-T-JEPA | $0.850\pm2.1\mathrm{e}^{-3}$ | $0.852\pm2.1\mathrm{e}^{-3}$ | $0.850\pm1.6\mathrm{e}^{-3}$ | $0.852\pm2.1\mathrm{e}^{-3}$ | $0.832\pm1.7\mathrm{e}^{-3}$ | $0.849\pm3.2\mathrm{e}^{-3}$ | $0.851\pm2.7\mathrm{e}^{-3}$ |
| MLP + Var-T-JEPA (10%) | $0.864\pm1.1\mathrm{e}^{-3}$ | $0.865\pm2.0\mathrm{e}^{-3}$ | $0.867\pm1.6\mathrm{e}^{-3}$ | $0.865\pm2.0\mathrm{e}^{-3}$ | $0.843\pm1.7\mathrm{e}^{-3}$ | $0.866\pm3.6\mathrm{e}^{-3}$ | $0.854\pm2.9\mathrm{e}^{-3}$ |
| MLP + Var-T-JEPA (20%) | $0.877\pm8.3\mathrm{e}^{-4}$ | $0.883\pm1.7\mathrm{e}^{-3}$ | $0.883\pm1.4\mathrm{e}^{-3}$ | $0.883\pm1.7\mathrm{e}^{-3}$ | $0.864\pm1.5\mathrm{e}^{-3}$ | $0.881\pm3.7\mathrm{e}^{-3}$ | $0.856\pm2.9\mathrm{e}^{-3}$ |
| MLP + Var-T-JEPA (50%) | $0.905\pm1.1\mathrm{e}^{-3}$ | $0.921\pm2.1\mathrm{e}^{-3}$ | $0.902\pm9.6\mathrm{e}^{-4}$ | $0.921\pm2.1\mathrm{e}^{-3}$ | $0.908\pm1.5\mathrm{e}^{-3}$ | $0.907\pm3.7\mathrm{e}^{-3}$ | $0.871\pm3.0\mathrm{e}^{-3}$ |
| ResNet + Var-T-JEPA | $0.852\pm1.2\mathrm{e}^{-3}$ | $0.854\pm1.4\mathrm{e}^{-3}$ | $0.851\pm6.5\mathrm{e}^{-4}$ | $0.854\pm1.4\mathrm{e}^{-3}$ | $0.833\pm1.5\mathrm{e}^{-3}$ | $0.853\pm8.7\mathrm{e}^{-4}$ | $0.852\pm2.0\mathrm{e}^{-3}$ |
| ResNet + Var-T-JEPA (10%) | $0.864\pm1.1\mathrm{e}^{-3}$ | $0.868\pm9.2\mathrm{e}^{-4}$ | $0.870\pm9.4\mathrm{e}^{-4}$ | $0.868\pm9.2\mathrm{e}^{-4}$ | $0.845\pm1.1\mathrm{e}^{-3}$ | $0.869\pm1.1\mathrm{e}^{-3}$ | $0.856\pm1.9\mathrm{e}^{-3}$ |
| ResNet + Var-T-JEPA (20%) | $0.877\pm1.4\mathrm{e}^{-3}$ | $0.886\pm8.2\mathrm{e}^{-4}$ | $0.885\pm9.9\mathrm{e}^{-4}$ | $0.886\pm8.2\mathrm{e}^{-4}$ | $0.865\pm1.2\mathrm{e}^{-3}$ | $0.884\pm2.0\mathrm{e}^{-3}$ | $0.858\pm1.6\mathrm{e}^{-3}$ |
| ResNet + Var-T-JEPA (50%) | $0.904\pm1.6\mathrm{e}^{-3}$ | $0.924\pm6.6\mathrm{e}^{-4}$ | $0.900\pm1.7\mathrm{e}^{-3}$ | $0.924\pm6.6\mathrm{e}^{-4}$ | $0.909\pm1.9\mathrm{e}^{-3}$ | $0.907\pm2.4\mathrm{e}^{-3}$ | $0.873\pm1.5\mathrm{e}^{-3}$ |

### B.3. Uncertainty Quantification: Comparison to Post-hoc Baselines

We further compare Var-T-JEPA's native latent uncertainty against four post-hoc estimators, all computed on the *same* frozen Var-T-JEPA embeddings: *Probe entropy* and *Probe MSP* (Hendrycks & Gimpel, 2017) (the predictive entropy and one minus the maximum softmax probability of a linear probe), *MC dropout* (Gal & Ghahramani, 2016) (predictive entropy of a dropout MLP over 20 forward passes), and *Deep ensemble* (Lakshminarayanan et al., 2017) (predictive entropy of five independently trained MLPs). On the controlled (semi-)synthetic SIM and MNIST datasets, which provide a ground-truth ambiguity score, we report the Spearman correlation with that score, the AUC for detecting high-ambiguity samples (i.e. those with simulated ambiguity score $u \geq q_{75}$, the 75th percentile), and the area under the risk–coverage curve (AURC). Table 6 shows that the native latent uncertainty attains the best Spearman and AUC on both datasets, while MC dropout and deep ensembles lead only on AURC.

*Table 6.* Uncertainty quantification: Var-T-JEPA's native latent uncertainty versus four post-hoc baselines on the same frozen embeddings, for the controlled SIM and MNIST datasets. Metrics: Spearman correlation with the ground-truth simulated ambiguity score, AUC for detecting high-ambiguity samples, and area under the risk–coverage curve (AURC).

| Dataset | Method | Spearman ↑ | AUC ↑ | AURC ↓ |
|---|---|---|---|---|
| | Var-T-JEPA (latent uncertainty) | **0.640** | **0.805** | 0.209 |
| | Probe entropy | 0.339 | 0.692 | 0.272 |
| **SIM** | Probe MSP | 0.313 | 0.668 | 0.267 |
| | MC dropout | 0.560 | 0.599 | 0.013 |
| | Deep ensemble | 0.586 | 0.612 | **0.011** |
| | Var-T-JEPA (latent uncertainty) | **0.949** | **0.986** | 0.123 |
| | Probe entropy | 0.476 | 0.773 | 0.054 |
| **MNIST** | Probe MSP | 0.462 | 0.764 | 0.054 |
| | MC dropout | 0.379 | 0.763 | **0.025** |
| | Deep ensemble | 0.444 | 0.782 | 0.028 |

## C. Dataset Descriptions

### C.1. Dataset Descriptions for Downstream Experiments with Var-T-JEPA

#### C.1.1. DATASET SUMMARY

Our experimental evaluation of Var-T-JEPA uses five real-world tabular datasets (Adult, Covertype, Electricity, Credit Card Default, Bank Marketing), a vision dataset (MNIST), where we treat every pixel as an individual feature column, and a fully synthetic dataset (SIM). The latter two datasets were adapted to exhibit controllable uncertainty (noise) in the original feature space. Table 7 summarizes their basic characteristics. For Covertype, we randomly subsample 10,000 examples for our experiments. For supervised downstream model training and evaluation we create train/validation/test splits (70/10/20) from the embeddings learned on the full feature dataset. For baseline models trained on raw features, we apply the same splits to the original datasets.

*Table 7.* Dataset characteristics used in experimental evaluation.

| Dataset | Samples | Features | (num/cat) | Classes | Task | Domain |
|---|---|---|---|---|---|---|
| Adult (AD) (Kohavi, 1996) | 48,842 | 14 | (6/8) | 2 | Classification | Census |
| Covertype (CO) (Blackard, 1998) | 10,000 | 54 | (54/0) | 7 | Classification | Forestry |
| Electricity (EL) (Harries, 1999) | 45,312 | 8 | (8/0) | 2 | Classification | Time series |
| Credit Card Default (CC) (Yeh, 2009) | 30,000 | 23 | (23/0) | 2 | Classification | Finance |
| Bank Marketing (BM) (Moro et al., 2014) | 45,211 | 16 | (16/0) | 2 | Classification | Marketing |
| MNIST (LeCun et al., 1998) | 10,000 | 784 | (784/0) | 10 | Classification | Vision |
| Simulated (SIM) | 10,000 | 32 | (28/4) | 3 | Classification | Synthetic |

### C.1.2. (SEMI-)SYNTHETIC DATA GENERATION

**Fully synthetic dataset (SIM).** Our simulated dataset is designed to produce well-separated class structure for most samples, while inducing controlled uncertainty in the original feature space for a subset of samples through prototype mixing. Concretely, we generate three class-conditional Gaussian clusters in a 28-dimensional numeric space with shared isotropic variance. We then draw a per-sample ambiguity score $(u_i \sim \mathrm{Uniform}(0,1))$ and map it to an amplified uncertainty/ambiguity strength $(u_i^{(\mathrm{amb})} = u_i^\gamma)$ with $(\gamma = 2)$. In doing so, most samples are easy and only a fraction become highly ambiguous. For each sample $(i)$ with true class $(c)$, we select an alternative class $(c')$ and blend the numeric features toward the alternative class prototype with strength $(\alpha_i \propto u_i^{(\mathrm{amb})})$. We further (i) mix a random subset of numeric features toward the alternative prototype, and (ii) inject additional ambiguity-driven noise in a small set of informative dimensions. Finally, we derive four categorical variables by quantile-binning selected numeric dimensions and flip categorical values with probability increasing in $u_i^{(\mathrm{amb})}$. The output includes 28 numeric features, 4 categorical features, and 3 classes.

**MNIST.** We represent MNIST as a tabular dataset by flattening each $28 \times 28$ grayscale image to a 784-dimensional feature vector and using the digit label as class to be predicted. To create a controllable uncertainty signal, we incorporate a per-sample score $(u_i \sim \mathrm{Uniform}(0,1))$ and corrupt the input by interpolating between the original image $(x_i)$ and an independent random image $(r_i \sim \mathrm{Uniform}(0,1)^{784})$:

$$\alpha_i = \mathrm{clip}(\lambda u_i, 0, 1), \qquad x_i' = (1 - \alpha_i)x_i + \alpha_i r_i,$$

with a fixed noise scale $(\lambda = 0.75)$. As illustrated in Figure 5, different values of $u_i$ produce varying levels of corruption, from clean images $(u_i = 0)$ to almost random noise $(u_i = 0.99)$. We use a random subset of 10,000 samples for the MNIST experiments.

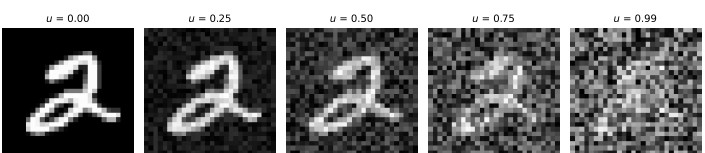

*Figure 5.* Visualization of MNIST image corruption under different values of the uncertainty score $u_i$.

### C.2. Data Generation for Simulation Study

We generate paired observations $(x_i, y_i) \in \mathbb{R}^{D_{\mathrm{obs}}} \times \mathbb{R}^{D_{\mathrm{obs}}}$ from latent variables $s_{x_i}, s_{y_i} \in \mathbb{R}^{d_s}$ and $z_i \in \mathbb{R}^{d_z}$. In all simulation experiments we use $D_{\mathrm{obs}} = 32$, $d_s = 16$, $d_z = 8$, diagonal covariances $\Sigma_x = \sigma_x^2 I_{d_s}$, $\Sigma_y = \sigma_y^2 I_{d_s}$, $\Sigma_z = \sigma_z^2 I_{d_z}$ with $(\sigma_x, \sigma_y, \sigma_z) = (1.0, 0.5, 1.0)$, a mean-shift $\delta = \delta_{\mathrm{scale}} \mathbf{1}$ with $\delta_{\mathrm{scale}} = 2.0$, and observation noise $(\tau_x, \tau_y) = (0.3, 0.3)$.

$$z_i \sim \mathcal{N}(0, \Sigma_z), \qquad s_{x_i} \sim \tfrac{1}{2}\mathcal{N}(0, \Sigma_x) + \tfrac{1}{2}\mathcal{N}(\delta, \Sigma_x), \qquad s_{y_i} \mid s_{x_i}, z_i \sim \mathcal{N}(s_{x_i} + Az_i, \Sigma_y),$$

$$x_i = h_x(s_{x_i}) + \epsilon_{x_i}, \; \epsilon_{x_i} \sim \mathcal{N}(0, \tau_x^2 I_{D_{\mathrm{obs}}}), \qquad y_i = h_y(s_{y_i}) + \epsilon_{y_i}, \; \epsilon_{y_i} \sim \mathcal{N}(0, \tau_y^2 I_{D_{\mathrm{obs}}}).$$

Equivalently, we introduce a binary mixture label $c_i \sim \mathrm{Bernoulli}(1/2)$ and sample $s_{x_i} \mid c_i \sim \mathcal{N}(c_i \delta, \Sigma_x)$. For each run, we draw $A \in \mathbb{R}^{d_s \times d_z}$ once with entries $A_{jk} \sim \mathcal{N}(0, 1/d_z)$ and keep it fixed. The nonlinear maps $h_x, h_y$ are also drawn once per run as independent two-layer MLPs with $\tanh$ activation (hidden width 64) and then frozen. Different runs use different draws of $(A, h_x, h_y)$ and therefore different simulated datasets.

# D. Implementation Details

## D.1. Var-JEPA Implementation for Simulation Study

The simulation study implements a minimal Var-JEPA variant to isolate the effects of per-sample KL regularization and aggregated-distribution regularization (SIGReg) under a ground-truth latent-variable data generator. We implement an MLP-based Var-JEPA with amortized Gaussian posteriors $q_\phi(s_x \mid x), q_\phi(z \mid s_x), q_\phi(s_y \mid s_x, z, y)$ and Gaussian decoders/conditional priors.

## D.2. Var-T-JEPA Implementation for Tabular Data

We provide implementation details for each module of Var-T-JEPA below, adopting standard components from T-JEPA (Thimonier et al., 2025) where appropriate.

### D.2.1. CONTEXT ENCODER IMPLEMENTATION

The context encoder $q_\phi(s_x^{(j)} \mid x)$ processes masked tabular input through three main stages:

**Feature tokenization.** Each feature $x^{(j)}$ from the masked input is embedded into a $D_{\text{hid}}$-dimensional token representation (in Var-T-JEPA we set $d = d_z = D_{\text{hid}}$). For numerical features, we use a shared linear projection with feature-specific bias terms $b^{(j)} \in \mathbb{R}^{D_{\text{hid}}}$ that account for different feature scales and distributions:

$$t_{\text{ctx}}^{(j)} = \begin{cases} W_{\text{num}} \cdot x^{(j)} + b^{(j)} + e_{\text{type}}^{\text{num}} + e_{\text{pos}}^{(j)} & \text{if } j \in \mathcal{I}_{\text{num}} \\ \text{Embed}_{L_j}(x^{(j)}) + e_{\text{type}}^{\text{cat}} + e_{\text{pos}}^{(j)} & \text{if } j \in \mathcal{I}_{\text{cat}} \end{cases} \tag{27}$$

where $W_{\text{num}} \in \mathbb{R}^{D_{\text{hid}} \times 1}$ is a shared linear projection for numerical features, $\text{Embed}_{L_j} : \{0, 1, \dots, L_j - 1\} \to \mathbb{R}^{D_{\text{hid}}}$ are learned embedding tables for categorical features with $L_j$ categories, $e_{\text{type}}$ denotes feature type embeddings distinguishing numerical from categorical features, and $e_{\text{pos}}^{(j)}$ denotes positional embeddings encoding feature position.

**Transformer processing.** The tokenized features are processed by a standard transformer encoder architecture (Vaswani et al., 2017) with multi-head self-attention and feed-forward layers. Following T-JEPA conventions, we prepend $N_{\text{CLS}}$ learnable classification tokens to the sequence, yielding contextualized representations:

$$h_{\text{ctx}} = \text{Transformer}([\{t_{\text{CLS}}^{(i)}\}_{i=1}^{N_{\text{CLS}}}, \{t_{\text{ctx}}^{(j)}\}_{j \in m_{\text{ctx}}}]) \in \mathbb{R}^{(N_{\text{CLS}} + M_{\text{ctx}}) \times D_{\text{hid}}} \tag{28}$$

**Variational output.** The context encoder outputs distributional parameters for the variational posterior:

$$\mu_\phi^{s_x}(h_{\text{ctx}}) = \text{Linear}(h_{\text{ctx}}) \tag{29}$$

$$\log \Sigma_\phi^{s_x}(h_{\text{ctx}}) = \text{Linear}(h_{\text{ctx}}) \tag{30}$$

$$q_\phi(s_x^{(j)} \mid x) = \mathcal{N}(s_x^{(j)}; \mu_\phi^{s_x}(h_{\text{ctx}})^{(j)}, \Sigma_\phi^{s_x}(h_{\text{ctx}})^{(j)}) \tag{31}$$

### D.2.2. AUXILIARY ENCODER IMPLEMENTATION

Both the auxiliary encoder and the target posterior use attention-based pooling to map the variable-length context sequence $s_x$ to a fixed set of pooled context tokens $\bar{s}_x = \{\bar{s}_x^{(i)}\}_{i=1}^{L_{\text{pool}}}$. This mechanism is order-invariant and yields a fixed number of pooled tokens regardless of $M_{\text{ctx}}$.

The auxiliary encoder $q_\phi(z \mid s_x)$ takes the pooled context representation $\bar{s}_x$ and processes it through a compact MLP. Consistent with the JEPA constraint, the auxiliary encoder conditions only on the context representation $s_x$, not on target information:

$$h_{\text{aux}}^{(l)} = \text{GELU}(\text{LayerNorm}(\text{Linear}(h_{\text{aux}}^{(l-1)}))) \tag{32}$$

$$\mu_\phi^z, \log \sigma_\phi^z = \text{MLP}_\phi(\bar{s}_x) \tag{33}$$

$$q_\phi(z \mid s_x) = \mathcal{N}(z; \mu_\phi^z, \text{diag}(\sigma_\phi^z)^2) \tag{34}$$

The auxiliary encoder uses fewer layers than the main encoders to maintain $z$ as a lightweight predictive hint rather than a complex feature representation.

D.2.3. TARGET POSTERIOR IMPLEMENTATION

Within the target posterior $q_\phi(s_{y_k}^{(j)}|s_x, z, w; m_{\text{trg}}^{(k)})$, we apply attention-based pooling to the variable-length context sequence $s_x$ to obtain $L_{\text{pool}}$ pooled context tokens before concatenating with $z$ and all feature tokens. This approach maintains proper conditioning on context latents $s_x$ and auxiliary latent $z$ while processing complete raw data $w$.

**Pooled token preparation.** The pooled context and auxiliary latent are projected into the token embedding space with type embeddings:

$$t_{\bar{s}_x}^{(i)} = W_{s_x} \cdot \bar{s}_x^{(i)} + e_{\text{type}}^{\text{pooled-ctx}} \quad \text{for } i = 1, \dots, L_{\text{pool}} \tag{35}$$

$$t_z = W_z \cdot z + e_{\text{type}}^{\text{aux-latent}} \tag{36}$$

where $W_{s_x}, W_z \in \mathbb{R}^{D_{\text{hid}} \times D_{\text{hid}}}$ are learned projection matrices, and $e_{\text{type}}^{\text{pooled-ctx}}, e_{\text{type}}^{\text{aux-latent}} \in \mathbb{R}^{D_{\text{hid}}}$ are learnable type embeddings.

**Feature tokenization.** Each feature $w^{(j)}$ from the complete data is embedded using the same tokenization scheme as the context encoder:

$$t_{\text{feat}}^{(j)} = \begin{cases} W_{\text{num}} \cdot w^{(j)} + b^{(j)} + e_{\text{type}}^{\text{num}} + e_{\text{pos}}^{(j)} & \text{if } j \in \mathcal{I}_{\text{num}} \\ \text{Embed}_{L_j}(w^{(j)}) + e_{\text{type}}^{\text{cat}} + e_{\text{pos}}^{(j)} & \text{if } j \in \mathcal{I}_{\text{cat}} \end{cases} \tag{37}$$

**Fixed-size augmented sequence processing.** The target posterior constructs a fixed-length augmented token sequence by concatenating CLS, pooled context, auxiliary, and feature tokens:

$$\text{seq}_{\text{aug}} = [\{t_{\text{CLS}}^{(i)}\}_{i=1}^{N_{\text{CLS}}}, \{t_{\bar{s}_x}^{(i)}\}_{i=1}^{L_{\text{pool}}}, t_z, \{t_{\text{feat}}^{(j)}\}_{j=1}^{D_{\text{feat}}}] \tag{38}$$

$$h_{\text{aug}} = \text{Transformer}(\text{seq}_{\text{aug}}) \tag{39}$$

$$\mu_\phi^{s_w}(h_{\text{aug}}), \log \Sigma_\phi^{s_w}(h_{\text{aug}}) = \text{Linear}(h_{\text{aug}}), \text{Linear}(h_{\text{aug}}) \tag{40}$$

The target latent distributions are extracted from the feature portion of the augmented sequence:

$$q_\phi(s_{y_k}^{(j)}|s_x, z, w; m_{\text{trg}}^{(k)}) = \mathcal{N}(s_{y_k}^{(j)}; \mu_\phi^{s_w}(h_{\text{feat}})^{(j)}, \Sigma_\phi^{s_w}(h_{\text{feat}})^{(j)}) \text{ for } j \in m_{\text{trg}}^{(k)} \tag{41}$$

where $h_{\text{feat}}$ denotes the feature token representations extracted from $h_{\text{aug}}$.

D.2.4. PREDICTIVE MODEL IMPLEMENTATION

The predictive model $p_\theta(s_{y_k}^{(j)}|s_x, z; m_{\text{trg}}^{(k)})$ operates entirely in latent space using learnable mask tokens. Unlike the target posterior, it has no access to raw target features and must predict latent representations from context and auxiliary information:

$$h_{\text{ctx}} = \text{Linear}(s_x) + \text{Linear}(z) + e_{\text{pos}}^{\text{ctx}} \tag{42}$$

$$h_{\text{mask}} = \text{MaskToken} + \text{Linear}(z) + e_{\text{pos}}^{\text{trg}} \tag{43}$$

$$h_{\text{pred}} = \text{Transformer}(\text{concat}(h_{\text{ctx}}, h_{\text{mask}})) \tag{44}$$

$$\mu_\theta^{s_y}(h_{\text{pred}}), \Sigma_\theta^{s_y}(h_{\text{pred}}) = \text{Linear}(h_{\text{pred}}), \text{Linear}(h_{\text{pred}}) \tag{45}$$

$$p_\theta(s_{y_k}^{(j)}|s_x, z; m_{\text{trg}}^{(k)}) = \mathcal{N}(s_{y_k}^{(j)}; \mu_\theta^{s_y}(h_{\text{pred}})^{(j)}, \Sigma_\theta^{s_y}(h_{\text{pred}})^{(j)}) \tag{46}$$

The predictive model does not directly distinguish between categorical and numerical features since it works purely in the latent representation space.

The predictor generates sequential target predictions for each target mask, enabling the model to handle multiple prediction tasks simultaneously during training. Crucially, this component has no access to target observations, maintaining the JEPA principle of predictive learning.

D.2.5. CONTEXT AND TARGET DECODER IMPLEMENTATION

The context and target decoders reconstruct original tabular features from their latent representations using feature-specific decoder heads. Both decoders share identical architectures but operate on different latent sequences.

**Feature-specific decoder architecture.** Each decoder maintains separate neural network heads for every original feature in the dataset:

$$\text{NumericDecoders} = \{u_\theta^{(j)} : \mathbb{R}^{D_{\text{hid}}} \to \mathbb{R}^1 \mid j \in \mathcal{I}_{\text{num}}\} \tag{47}$$

$$\text{CategoricalDecoders} = \{u_\theta^{(j)} : \mathbb{R}^{D_{\text{hid}}} \to \mathbb{R}^{C_j} \mid j \in \mathcal{I}_{\text{cat}}\} \tag{48}$$

where $C_j$ denotes the cardinality of categorical feature $j$. Each decoder head is implemented as a linear layer specific to the feature type and index.

**Context decoder processing.** Given a context latent sequence $s_x \in \mathbb{R}^{M_{\text{ctx}} \times D_{\text{hid}}}$ and context feature indices $m_{\text{ctx}} \in \mathbb{R}^{M_{\text{ctx}}}$, the context decoder processes each latent token:

$$\text{for } t = 1, \ldots, M_{\text{ctx}}: \quad j = m_{\text{ctx}}[t], \quad \hat{x}^{(j)} = u_\theta^{(j)}(s_x^{(t)}) \tag{49}$$

**Target decoder processing.** The target decoder follows identical processing and operates directly on target latent sequences:

$$\hat{w}^{(j)} = u_\theta^{(j)}(s_w^{(j)}) \tag{50}$$

**Reconstruction distributions.** The decoder outputs are interpreted as distributional parameters for any observation $\pi \in \{x, w\}$ with corresponding latent representation $s_\pi$:

$$p_\theta(\pi^{(j)} | s_\pi^{(j)}) = \begin{cases} \mathcal{N}(\pi^{(j)}; u_\theta^{(j)}(s_\pi^{(j)}), \sigma^2) & \text{if } j \in \mathcal{I}_{\text{num}} \\ \text{Categorical}(\pi^{(j)}; \text{softmax}(u_\theta^{(j)}(s_\pi^{(j)}))) & \text{if } j \in \mathcal{I}_{\text{cat}} \end{cases} \tag{51}$$

where $\sigma^2$ is a learnable global noise parameter for numerical reconstruction uncertainty. The full reconstruction distribution factorizes as $p_\theta(\pi | s_\pi) = \prod_{j \in m_\pi} p_\theta(\pi^{(j)} | s_\pi^{(j)})$, where $m_\pi$ denotes the mask for observation $\pi$.

### D.2.6. KL DIVERGENCES

We compute the closed form KL divergences between Gaussians, which is given by (Murphy, 2022)

$$\text{KL}\big(\mathcal{N}(\mu_q, \Sigma_q) \,\|\, \mathcal{N}(\mu_p, \Sigma_p)\big) = \tfrac{1}{2}\Big[ +\text{tr}\big(\Sigma_p^{-1}\Sigma_q\big) + (\mu_p - \mu_q)^\top \Sigma_p^{-1}(\mu_p - \mu_q) + \log \frac{\det \Sigma_p}{\det \Sigma_q} - d \Big]. \tag{52}$$

In our implementation, all latent variables use diagonal covariance matrices $\Sigma = \text{diag}(\Sigma_1, \ldots, \Sigma_{D_{\text{hid}}})$ where each $\Sigma_d$ represents the variance of dimension $d$. As our priors have $\mu_p = 0$, $\Sigma_p = I$ for $s_x$ and $z$, Eq. (52) simplifies to:

$$\text{KL}\big(q_\phi(s_x^{(j)} | x) \,\|\, \mathcal{N}(0, I)\big) = \tfrac{1}{2} \sum_{d=1}^{D_{\text{hid}}} \Big[ (\Sigma_\phi^{s_x^{(j,d)}}) + (\mu_\phi^{s_x^{(j,d)}})^2 - \log(\Sigma_\phi^{s_x^{(j,d)}}) - 1 \Big]$$

$$\text{KL}\big(q_\phi(z | s_x) \,\|\, \mathcal{N}(0, I)\big) = \tfrac{1}{2} \sum_{d=1}^{D_{\text{hid}}} \Big[ (\Sigma_\phi^{z^{(d)}}) + (\mu_\phi^{z^{(d)}})^2 - \log(\Sigma_\phi^{z^{(d)}}) - 1 \Big]$$

The third KL term between two diagonal Gaussians becomes:

$$\text{KL}\big(q_\phi(s_{y_k}^{(j)} | s_x, z, w; m_{\text{trg}}^{(k)}) \,\|\, p_\theta(s_{y_k}^{(j)} | s_x, z; m_{\text{trg}}^{(k)})\big) = \tfrac{1}{2} \sum_{d=1}^{D_{\text{hid}}} \Big[ \frac{(\Sigma_\phi^{s_{y_k}^{(j,d)}})}{(\Sigma_\theta^{s_{y_k}^{(j,d)}})} + \frac{(\mu_\theta^{s_{y_k}^{(j,d)}} - \mu_\phi^{s_{y_k}^{(j,d)}})^2}{(\Sigma_\theta^{s_{y_k}^{(j,d)}})} + \log \frac{(\Sigma_\theta^{s_{y_k}^{(j,d)}})}{(\Sigma_\phi^{s_{y_k}^{(j,d)}})} - 1 \Big]$$

where $\mu_\phi^{s_{y_k}^{(j,d)}} = \mu_\phi^{s_w^{(j,d)}} : j \in m_{\text{trg}}^{(k)}$ and $\Sigma_\phi^{s_{y_k}^{(j,d)}} = \Sigma_\phi^{s_w^{(j,d)}} : j \in m_{\text{trg}}^{(k)}$ are the masked means and covariances for target $y_k$.

### D.3. Training and Testing Procedure

This section provides detailed implementation specifics for training the Var-T-JEPA model, including the training algorithm and KL annealing schedules.

### D.3.1. TRAINING ALGORITHM

Algorithm 1 outlines the complete training procedure for our Var-T-JEPA for tabular data. Note that, while we used a per-sample notation in the main paper, the algorithm presents the batch-level implementation where loss terms represent averages over mini-batches and variables are indexed by sample position within the batch.

---

**Algorithm 1:** Var-T-JEPA Training Procedure

---

**Input:** Dataset $\mathcal{D}$, mask parameters $(r_{\text{ctx}}^{\min}, r_{\text{ctx}}^{\max}, r_{\text{trg}}^{\min}, r_{\text{trg}}^{\max})$, number of target views $K$ (and typically 1 context view)

**Input:** KL annealing schedules over optimizer steps $\{\alpha_{s_x}^{\text{KL}}(t), \alpha_z^{\text{KL}}(t), \alpha_{s_y}^{\text{KL}}(t)\}$

1 **while** *not converged and epoch < max epochs* **do**

2    **for** *each batch in dataloader* **do**

3      Sample batch feature vectors $\{w^{(n)}\}_{n=1}^B \sim \mathcal{D}$;

4      **Generate masks (MaskCollator):**;

5      Sample mask sizes $M_{\text{ctx}}, M_{\text{trg}}$ uniformly from the configured ranges, then generate context masks $\{m_{\text{ctx}}^{(n)}\}_{n=1}^B$ and $K$ target masks $\{m_{\text{trg}}^{(k,n)}\}_{k=1}^K$ per sample;

6      Get current KL weights from schedules at global optimizer-step $t$: $\alpha_{s_x}^{\text{KL}}(t), \alpha_z^{\text{KL}}(t), \alpha_{s_y}^{\text{KL}}(t)$;

7      **Forward Pass (vectorized over target masks):**;

8      Sample $\{s_x^{(n)} \sim q_\phi(s_x \mid w^{(n)}; m_{\text{ctx}}^{(n)})\}_{n=1}^B$ using reparameterization trick;

9      Sample $\{z^{(n)} \sim q_\phi(z \mid s_x^{(n)})\}_{n=1}^B$ using a pooled context representation;

10      Concatenate target masks over $k$ to form an effective target batch of size $B \cdot K$; repeat $s_x, z$, and $w$ to match this effective batch;

11      Sample target posterior latents for masked target features: $s_{y_k}^{(n)} \sim q_\phi(s_y \mid s_x^{(n)}, z^{(n)}, w^{(n)}; m_{\text{trg}}^{(k,n)})$;

12      Predict target latents for the same masked features: $p_\theta(s_{y_k}^{(n)} \mid s_x^{(n)}, z^{(n)}; m_{\text{trg}}^{(k,n)})$;

13      **Compute Losses:**;

14      $\mathcal{L}^{\text{rec}} = \frac{1}{B} \sum_{n=1}^B \frac{1}{M_{\text{ctx}}} \sum_{j \in m_{\text{ctx}}^{(n)}} -\log p_\theta(x^{(j,n)} | s_x^{(j,n)})$;

15      $\mathcal{L}^{\text{gen}} = \frac{1}{B \cdot K} \sum_{n=1}^B \sum_{k=1}^K \frac{1}{D_{\text{feat}}} \sum_{j=1}^{D_{\text{feat}}} -\log p_\theta(w^{(j,n)} \mid s_w^{(j,n)})$;

16      $\mathcal{L}_{s_x}^{\text{KL}} = \frac{1}{B} \sum_{n=1}^B \frac{1}{M_{\text{ctx}}} \sum_{j \in m_{\text{ctx}}^{(n)}} \text{KL}(q_\phi(s_x^{(j,n)} | x^{(n)}) \| \mathcal{N}(0, I))$;

17      $\mathcal{L}_z^{\text{KL}} = \frac{1}{B} \sum_{n=1}^B \text{KL}(q_\phi(z^{(n)} | s_x^{(n)}) \| \mathcal{N}(0, I))$;

18      $\mathcal{L}_{s_y}^{\text{KL}} =$

        $\frac{1}{B \cdot K} \sum_{n=1}^B \sum_{k=1}^K \frac{1}{M_{\text{trg}}} \sum_{j \in m_{\text{trg}}^{(k,n)}} \text{KL}(q_\phi(s_{y_k}^{(j,n)} | s_x^{(n)}, z^{(n)}, w^{(n)}; m_{\text{trg}}^{(k,n)}) \| p_\theta(s_{y_k}^{(j,n)} | s_x^{(n)}, z^{(n)}; m_{\text{trg}}^{(k,n)}))$;

19      $\mathcal{L} = \alpha^{\text{rec}} \mathcal{L}^{\text{rec}} + \alpha^{\text{gen}} \mathcal{L}^{\text{gen}} + \alpha_{s_x}^{\text{KL}}(t) \mathcal{L}_{s_x}^{\text{KL}} + \alpha_z^{\text{KL}}(t) \mathcal{L}_z^{\text{KL}} + \alpha_{s_y}^{\text{KL}}(t) \mathcal{L}_{s_y}^{\text{KL}}$;

20      Compute gradients: $\nabla_{\theta, \phi} \mathcal{L}$;

21      Update parameters using optimizer (e.g., AdamW);

22      Increment KL annealing step: $t \leftarrow t + 1$;

23    Optionally run downstream probe every $C$ epochs and update best checkpoint based on validation score;

24    **if** *no downstream improvement for $P$ probe evaluations* **then**

25      Load best checkpoint and terminate training;

---

### D.3.2. KL DIVERGENCE ANNEALING

The KL divergence terms in the ELBO are gradually introduced during training through separate annealing schedules for each latent variable type. This prevents posterior collapse and ensures stable training dynamics, a common technique in training variational autoencoders (Bowman et al., 2016). We implement linear annealing schedules with configurable start times and durations:

$$\alpha_{s_x}^{\text{KL}}(t) = \min\left(\frac{t}{T_{s_x}^{\text{anneal}}}, 1\right) \cdot \alpha_{s_x, \text{final}}^{\text{KL}} \tag{53}$$

$$\alpha_z^{\text{KL}}(t) = \min\left(\frac{t}{T_z^{\text{anneal}}}, 1\right) \cdot \alpha_{z,\text{final}}^{\text{KL}} \tag{54}$$

$$\alpha_{s_y}^{\text{KL}}(t) = \min\left(\frac{t}{T_{s_y}^{\text{anneal}}}, 1\right) \cdot \alpha_{s_y,\text{final}}^{\text{KL}} \tag{55}$$

where $t$ denotes the current training step, $T_{\cdot}^{\text{anneal}}$ specifies the annealing duration, and $\alpha_{\cdot,\text{final}}^{\text{KL}}$ sets the final weight values.

### D.4. T-JEPA Implementation for Tabular Data

T-JEPA (Thimonier et al., 2025) employs a deterministic joint-embedding architecture with three main components: context encoder $f_\phi^{\text{ctx}}$, target encoder $f_\phi^{\text{trg}}$, and predictor $g_\theta$, as illustrated in Figure 6.

**Context and target encoders.** Both encoders $f_\phi^{\text{ctx}}$ and $f_\phi^{\text{trg}}$ share identical architectures but operate on different masked views of the input. T-JEPA uses the same tokenization scheme as described in Eq. (27) and Eq. (37), processing tabular features through feature-specific embeddings with type and positional encodings. The tokenized features are processed by a transformer encoder to produce deterministic representations $s_x = f_\phi^{\text{ctx}}(x)$ and $s_y = f_\phi^{\text{trg}}(w; m_{\text{trg}})$ for context and target views respectively.

**Predictor.** As shown in Figure 6, the predictor $g_\theta$ learns to forecast target representations from context representations. The predictor is implemented using a similar transformer architecture as our Var-T-JEPA, including learnable mask tokens with positional embeddings, used to predict target features $\hat{s}_y = g_\theta(s_x; m_{\text{trg}})$.

**Training objective.** T-JEPA minimizes the averaged mean squared error between predicted and actual target representations over all target masks and features:

$$\mathcal{L}_{\text{T-JEPA}} = \frac{1}{K \cdot M_{\text{trg}}} \sum_{k=1}^{K} \sum_{j \in m_{\text{trg}}^{(k)}} \|\hat{s}_{y_k}^{(j)} - s_{y_k}^{(j)}\|^2 \tag{56}$$

The target encoder parameters are updated via exponential moving average (EMA) of the context encoder parameters to ensure stable training, while the predictor is trained end-to-end with gradient descent.

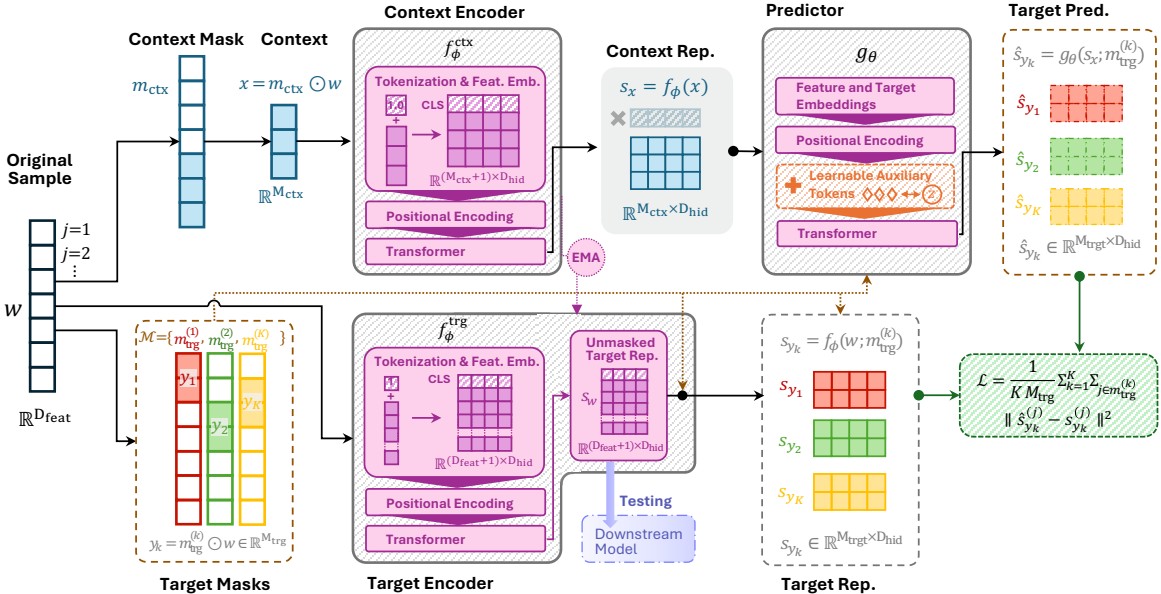

*Figure 6.* Architectural overview of the T-JEPA pipeline by Thimonier et al. (2025) extended by learnable auxiliary tokens.

**Learnable auxiliary tokens.** The original T-JEPA framework by Thimonier et al. (2025) does not incorporate the auxiliary variable $z$ in its predictor architecture. To match our theoretical formulation where predictions depend on both context representations $s_x$ and auxiliary variables $z$, we introduce $N_{\text{aux}}$ learnable auxiliary tokens $\text{AuxToken} \in \mathbb{R}^{N_{\text{aux}} \times D_{\text{hid}}}$ as trainable parameters that serve as dedicated slots for auxiliary predictive information. These tokens are augmented with distinct positional embeddings $e_{\text{pos}}^{\text{aux}} \in \mathbb{R}^{N_{\text{aux}} \times D_{\text{hid}}}$ and concatenated with context and target tokens in the predictor input sequence:

$$\text{Concat}(s_x, \text{AuxToken} + e_{\text{pos}}^{\text{aux}}, \text{MaskToken} + e_{\text{pos}}^{\text{trg}}) \rightarrow \text{Transformer} \rightarrow \hat{s}_{y_k} \tag{57}$$

where $\text{MaskToken} + e_{\text{pos}}^{\text{trg}}$ denotes the masked target tokens with their positional embeddings, and $\hat{s}_{y_k}$ represents the predicted target latent representation. The auxiliary tokens provide learnable parameters that can encode predictive hints analogous to sampling from $q_\phi(z|s_x)$ in our variational framework, enabling the deterministic T-JEPA predictor to access auxiliary predictive capacity while maintaining end-to-end differentiability.

### D.5. Embedding Generation and Uncertainty Estimates for Downstream Evaluation

For downstream evaluation, our tabular Var-T-JEPA deviates from the general test-time procedure described in Section 3.6 in two key aspects. First, the target posterior $q_\phi(s_{y_k}^{(j)}|s_x, z, w; m_{\text{trg}}^{(k)})$ conditions on the feature vector $w$ since tabular features are accessible during embedding generation, unlike the general Var-JEPA framework where target observations may generally be unavailable. Second, the auxiliary latent is generated via the learned auxiliary encoder $q_\phi(z|s_x)$ rather than sampling from the standard normal prior $p(z) = \mathcal{N}(0, I)$, maintaining consistency with the training regime where the auxiliary encoder learns meaningful context-to-auxiliary mappings for downstream representation quality. Following standard practice in representation learning, final embeddings for downstream analysis were generated using distributional means rather than sampling to ensure deterministic representations: $s_x = \mu_\phi^{s_x}$, $z = \mu_\phi^z$, and $s_w = \mu_\phi^{s_w}$. For our downstream evaluation experiments of both our tabular Var-T-JEPA and T-JEPA, we use the target posterior representations $s_w$ as the final embeddings, which capture the joint information from context, auxiliary latents, and complete feature observations. Uncertainty estimates for selective evaluation are computed by aggregating the standard deviations of the target latent posterior distribution: for real-world datasets (AD, CO, EL, CC, BM) we use mean aggregation, while for (semi-)synthetic datasets (MNIST, SIM) with known ground-truth uncertainty we use the 90th percentile to better capture tail behavior. For embedding and uncertainty extraction, checkpoints are selected based on online linear-probe validation performance: for T-JEPA we use the best unfiltered validation accuracy, while for Var-T-JEPA we use the best filtered validation accuracy with the most-uncertain $20\%$ of validation samples discarded.

### D.6. Hyperparameter Settings

#### D.6.1. COMPUTE RESOURCES

Var-T-JEPA and T-JEPA were trained on a computer cluster with one NVIDIA A100 GPU (40GB or 80GB VRAM), two CPU cores, and 20 GB system RAM per task. Var-T-JEPA training time varied by dataset and configuration, and was typically between under 2 hours and less than 24 hours per run across our experiments.

#### D.6.2. VAR-T-JEPA HYPERPARAMETERS

Table 8 summarizes the hyperparameters used for the tabular Var-T-JEPA experiments. Notation: 'batch': batch size; 'lr': learning rate; 'warm': warmup epochs; 'ctx'/'ctxM': min/max context mask share; 'trg'/'trgM': min/max target mask share; 'K': number of target predictions; 'd': hidden dimension; 'L': number of layers; 'H': number of heads; 'ff': feedforward dimension; 'drop': dropout probability; 'Lp': predictor embedding dimension; 'Hp': predictor number of heads; 'ffp': predictor feedforward dimension; 'dropp': predictor dropout probability; '$\alpha_{s_x}^{\text{KL}}$', '$\alpha_z^{\text{KL}}$', '$\alpha_{s_y}^{\text{KL}}$': final KL weights for $s_x$, $z$, $s_y$; '$T_{s_x}^{\text{anneal}}$', '$T_z^{\text{anneal}}$', '$T_{s_y}^{\text{anneal}}$': KL annealing epochs for $s_x$, $z$, $s_y$; '$\alpha^{\text{rec}}$', '$\alpha^{\text{gen}}$': reconstruction weights for $x$, $y$.

*Table 8.* Var-T-JEPA hyperparameters used for downstream experiments.

| Dataset | batch | lr | warm | ctx | ctxM | trg | trgM | K | d | L | H | ff | drop | Lp | Hp | ffp | dropp | $\alpha_{s_x}^{\text{KL}}$ | $\alpha_z^{\text{KL}}$ | $\alpha_{s_y}^{\text{KL}}$ | $T_{s_x}^{\text{anneal}}$ | $T_z^{\text{anneal}}$ | $T_{s_y}^{\text{anneal}}$ | $\alpha^{\text{rec}}$ | $\alpha^{\text{gen}}$ |
|---|---|---|---|---|---|---|---|---|---|---|---|---|---|---|---|---|---|---|---|---|---|---|---|---|---|
| AD | 512 | 0.001 | 10 | 0.1 | 0.3 | 0.1 | 0.6 | 4 | 64 | 8 | 4 | 256 | 0.001 | 16 | 4 | 256 | 0.002 | 1e-04 | 1e-06 | 1e-05 | 15 | 15 | 15 | 0.1 | 1 |
| CO | 512 | 5e-04 | 12 | 0.15 | 0.35 | 0.15 | 0.6 | 4 | 64 | 8 | 8 | 64 | 0.0015 | 16 | 4 | 256 | 0.002 | 1e-06 | 1e-05 | 1e-06 | 60 | 60 | 20 | 0.001 | 0.5 |
| EL | 512 | 5e-04 | 10 | 0.15 | 0.6 | 0.15 | 0.9 | 4 | 64 | 6 | 4 | 64 | 0.001 | 16 | 4 | 128 | 0.002 | 1e-06 | 1e-05 | 1e-06 | 40 | 40 | 15 | 0.001 | 0.25 |
| CC | 512 | 0.001 | 10 | 0.1 | 0.3 | 0.1 | 0.6 | 4 | 64 | 8 | 4 | 256 | 0.001 | 16 | 4 | 256 | 0.002 | 1e-04 | 1e-06 | 1e-05 | 15 | 15 | 15 | 0.1 | 1 |
| BM | 512 | 0.001 | 10 | 0.1 | 0.3 | 0.1 | 0.6 | 4 | 64 | 8 | 4 | 256 | 0.001 | 16 | 4 | 256 | 0.002 | 1e-04 | 1e-06 | 1e-05 | 15 | 15 | 15 | 0.1 | 1 |
| MNIST | 256 | 0.001 | 10 | 0.15 | 0.5 | 0.15 | 0.8 | 4 | 64 | 8 | 4 | 128 | 0.002 | 32 | 4 | 256 | 0.002 | 1e-06 | 1e-06 | 1e-06 | 100 | 100 | 100 | 0.001 | 0.1 |
| SIM | 512 | 5e-04 | 10 | 0.15 | 0.5 | 0.15 | 0.8 | 4 | 64 | 16 | 2 | 64 | 0.002 | 16 | 4 | 256 | 0.002 | 1e-06 | 1e-06 | 1e-05 | 50 | 50 | 50 | 0.001 | 0.1 |

### D.6.3. T-JEPA HYPERPARAMETERS

Table 9 summarizes the hyperparameters used for the tabular T-JEPA experiments. For any hyperparameter that has a direct counterpart in Var-T-JEPA, we use the same setting as in Var-T-JEPA. Notation matches Var-T-JEPA: 'batch': batch size; 'lr': learning rate; 'warm': warmup epochs; 'ctx'/'ctxM': min/max context mask share; 'trg'/'trgM': min/max target mask share; 'K': number of target predictions; 'd': hidden dimension; 'L': number of layers; 'H': number of heads; 'ff': feedforward dimension; 'drop': dropout probability; 'Lp': predictor embedding dimension; 'Hp': predictor number of heads; 'ffp': predictor feedforward dimension; 'dropp': predictor dropout probability; 'ema': EMA decay rate.

*Table 9.* T-JEPA hyperparameters used for downstream experiments.

| Dataset | batch | lr | warm | ctx | ctxM | trg | trgM | K | d | L | H | ff | drop | Lp | Hp | ffp | dropp | ema |
|---|---|---|---|---|---|---|---|---|---|---|---|---|---|---|---|---|---|---|
| AD | 512 | 0.001 | 10 | 0.1 | 0.3 | 0.1 | 0.6 | 4 | 64 | 8 | 4 | 256 | 0.001 | 16 | 4 | 256 | 0.002 | 0.996 |
| CO | 512 | 5e-04 | 12 | 0.15 | 0.35 | 0.15 | 0.6 | 4 | 64 | 8 | 8 | 64 | 0.0015 | 16 | 4 | 256 | 0.002 | 0.996 |
| EL | 512 | 5e-04 | 10 | 0.15 | 0.6 | 0.15 | 0.9 | 4 | 64 | 6 | 4 | 64 | 0.001 | 16 | 4 | 128 | 0.002 | 0.996 |
| CC | 512 | 0.001 | 10 | 0.1 | 0.3 | 0.1 | 0.6 | 4 | 64 | 8 | 4 | 256 | 0.001 | 16 | 4 | 256 | 0.002 | 0.996 |
| BM | 512 | 0.001 | 10 | 0.1 | 0.3 | 0.1 | 0.6 | 4 | 64 | 8 | 4 | 256 | 0.001 | 16 | 4 | 256 | 0.002 | 0.996 |
| MNIST | 256 | 0.001 | 10 | 0.15 | 0.5 | 0.15 | 0.8 | 4 | 64 | 8 | 4 | 128 | 0.002 | 32 | 4 | 256 | 0.002 | 0.996 |
| SIM | 512 | 5e-04 | 10 | 0.15 | 0.5 | 0.15 | 0.8 | 4 | 64 | 16 | 2 | 64 | 0.002 | 16 | 4 | 256 | 0.002 | 0.996 |

### D.6.4. DOWNSTREAM AND BASELINE MODEL HYPERPARAMETERS

Downstream predictors were trained either on raw features or on learned embeddings. For embedding-based evaluation, embeddings are flattened for neural models. For raw-feature baselines, standard tabular encoders are used. All neural models (MLP, DCNv2, ResNet, AutoInt, FT-Transformer) share common hyperparameters: learning rate 0.001, weight decay 0.01, batch size 128, 50 training epochs with exponential patience of 16. Model-specific parameters include: MLP (2 hidden layers, hidden dimension 128, dropout 0.1); DCNv2 (2 cross layers); ResNet (3 blocks, output dimension 128, block dimension 128, hidden dimension 256, dropout 0.1); AutoInt (token dimension 128, 2 attention heads, 3 layers); FT-Transformer (standard transformer architecture). XGBoost uses a tree-based configuration with 100 estimators, max depth 3, learning rate 0.1, and does not use embedding flattening. We additionally use an online linear probe for downstream evaluation during Var-T-JEPA training and model selection, trained with learning rate $10^{-3}$, weight decay $2 \times 10^{-5}$, batch size 128, and 50 epochs. For the risk–coverage curves in Figure 3, we train a linear probe on standardized embeddings for 20 epochs with learning rate $10^{-2}$ and batch size 8192.

### D.6.5. SIMULATION STUDY (VAR-JEPA) HYPERPARAMETERS

We use batch size 512. Training runs for 40 epochs with learning rate $10^{-3}$ and weight decay $10^{-6}$. The Var-JEPA MLP uses hidden dimension 128 and depth 2. Default loss weights are $\alpha^{\mathrm{rec}} = \alpha^{\mathrm{gen}} = \alpha^{\mathrm{KL}}_{s_x} = \alpha^{\mathrm{KL}}_{z} = \alpha^{\mathrm{KL}}_{s_y} = 1$. SIGReg uses 64 directions, 64 frequencies, maximum frequency 5.0, with strengths $\lambda_{s_x} = 10$ and $\lambda_{s_y} = 10$. Linear probes are trained for 50 epochs with learning rate $3 \times 10^{-3}$, weight decay $10^{-4}$, batch size 512, and hidden dimension 64.

