# OpenReview forum: "Var-JEPA: A Variational Formulation of the Joint-Embedding Predictive Architecture – Bridging Predictive and Generative Self-Supervised Learning"
_ICML.cc/2026/Conference — ICML 2026 regular_

### Official Review · Reviewer_18NP · 2026-03-01

**Soundness:** 2
**Presentation:** 3
**Significance:** 2
**Originality:** 2
**Overall Recommendation:** 4
**Confidence:** 3

**Summary:**

This paper proposes Var-JEPA, a variational reformulation of the Joint-Embedding Predictive Architecture (JEPA). In Var-JEPA, the predictor becomes a learned conditional prior over target latents, and the overall training objective is a single ELBO, which explicitly includes reconstruction/generation terms and KL regularization. This formulation is claimed to (i) reduce reliance on ad-hoc anti-collapse heuristics, and (ii) enable principled uncertainty quantification through posterior covariances. The work further instantiates the approach for heterogeneous tabular data as Var-T-JEPA, and reports a simulation ablation study plus downstream tabular evaluations showing improved robustness compared to T-JEPA and competitive performance versus strong raw-feature baselines.

**Compliance With Llm Reviewing Policy:**

Affirmed.

**Final Justification:**

Fully addressed

**Key Questions For Authors:**

1. Given that JEPA has been primarily impactful in vision and video domains, why did authors choose to focus on tabular data for Var-JEPA experiments? Are there technical obstacles that prevent a straightforward extension to images?
2. Table 2 shows that after introducing var-JEPA, the performance improvements vary significantly across different experimental settings. Could authors further analyze how the introduction of Var-JEPA adapts to different data characteristics and uncertainties?
3. In the experiments, do authors ever observe representation collapse on sy (e.g., very small covariance, low mutual information with y)? If yes, how mitigate it?
4. Have authors evaluated whether the added decoders and latent distributions significantly increase training time or memory usage compared to T-JEPA?

**Limitations:**

No, the paper should more explicitly acknowledge the restriction to tabular domains and discuss implications for higher-dimensional settings.

**Strengths And Weaknesses:**

Strengths:
1.	The conceptual reframing of JEPA as a deterministic specialization of a coupled VAE with a learned conditional prior feels genuinely insightful. The explicit comparison to LeJEPA and SIGReg, and the demonstration that ELBO’s per sample KL often suffices to shape aggregated distribution and provides novel empirical insights.
2.	The exposition is clear and pedagogical; the paper is easy to follow even for readers not deeply familiar with JEPA internals. The figures and table layouts are informative, especially the diagnostic plots showing how distribution metrics evolve across training and variants.
3.	The simulation study is well designed to isolate the effects of different ELBO components (reconstruction vs KL, KL vs SIGReg), supporting the claim that per-sample KL terms can enforce reasonable aggregated distributions without separate SIGReg.
4.	The selective prediction experiments in tabular domains are compelling: they demonstrate that learned uncertainty is not just a byproduct but an actionable signal.


Weaknesses:
1.	The relationship between the data characteristics of different domains and the importance of uncertainty is not sufficiently elaborated. More ablations on uncertainty calibration (e.g., comparison to other uncertainty methods) would strengthen the soundness of the uncertainty-related claims.
2.	Some notation (e.g., distinguishing training-time vs inference-time use of Q_phi and P_theta) could be emphasized more explicitly to avoid confusion for less experienced readers.
3.	The immediate impact is clearest in tabular and “JEPA theory” circles; the broader community working on large-scale vision/video JEPA might still withhold judgment until they see largescale experiments.
4.	From an algorithmic viewpoint, the method is a natural extension of existing VAE machinery to JEPA; some might view the contribution as incremental unless the practical advantages are more strongly demonstrated.

---

> ### Author Rebuttal · Authors · 2026-03-29
>
> Dear Reviewer,
>
> thank you for the thoughtful assessment. We would like to briefly emphasize our paper's main message again, as it may otherwise be easy to miss:
> ___
> ***Why Var-JEPA changes the perspective on JEPA***
>
> *Our central claim goes beyond proposing a stronger JEPA variant: the predictor-encoder structure of JEPA already admits a latent-variable interpretation, making the divide between JEPA and probabilistic generative modeling less fundamental than it appears. Cast in a variational formulation, the same structure becomes conceptually richer, empirically more robust, and yields principled uncertainty estimation – framing standard JEPA as a restricted deterministic instance of a broader variational construction rather than a separate paradigm. We believe this insight – clarifying a widely-held but incomplete picture of what JEPA is really doing – is relevant to the SSL community and deserves to be showcased at ICML.*
> ___
>
> **1. Why tabular data, and are there technical obstacles to images?**
>
> We chose tabular data to isolate the objective-level contribution cleanly: image and video JEPA systems are dominated by architectural scale, tokenization, masking, and optimization choices that confound attribution to the variational objective. The tabular setting lets us examine collapse, latent geometry, and uncertainty (Tables 1–2, Figures 2–3). Our claim is not that tabular is the final target – once JEPA is written as a coupled latent-variable model with a learned conditional prior, this opens a broader design space for image and video. A direct port of I-JEPA/V-JEPA would otherwise inherit design choices tuned for deterministic objectives.
>
> **2. Why do improvements vary across datasets?**
>
> The non-uniform gains are meaningful rather than contradictory: Var-JEPA helps most where the context-to-target mapping is stochastic or underdetermined – exactly where latent regularization and uncertainty modeling matter – while deterministic settings naturally yield smaller margins (Table 2). Figure 3 supports this: learned latent uncertainty tracks simulated ambiguity. The results suggest the probabilistic formulation is most valuable when data ambiguity is real, not a universal claim.
>
> For hyperparameter sensitivity across datasets, we refer to our reply (3) to Reviewer `e7io`, where we report the corresponding multi-dataset sensitivity results.
>
> **3. Did we observe collapse on $s_y$?**
>
> Table 1 confirms that $s_y$ can collapse when key ELBO terms are removed. The clearest case is variant G (no reconstruction/generation terms), where probe accuracy drops from 0.993 to 0.543 and covariance diagnostics contract, indicating $s_y$ no longer carries rich information. Since $s_y$ is regularized toward the learned conditional prior $p_\theta(s_y \mid s_x,z)$ rather than $\mathcal{N}(0,I)$, isotropy is not the right collapse criterion. The complete ELBO thus provides principled regularization for $s_x$ and $s_y$; incomplete objectives do lead to collapse.
>
> **4. Training memory overhead**
>
> We additionally benchmarked training overhead on the Adult dataset (100 epochs). Var-T-JEPA increases GPU memory usage, as expected from the additional probabilistic components, but training time remains comparable to T-JEPA in wall-clock terms, potentially partly reflecting the extra target-encoder/EMA update path in T-JEPA.
>
> |Method|Epochs|Avg sec/epoch|Train time (sec)|Peak GPU memory (MB)|Params|
> |-|-|-|-|-|-|
> |T-JEPA|100|13.934|1393.380|1665.1|1092704|
> |Var-T-JEPA|100|13.001|1300.136|2966.5|1163146|
>
> **5. Additional uncertainty ablations**
>
> We compared Var-T-JEPA's native latent uncertainty against post-hoc baselines (probe entropy, MSP, MC dropout, deep ensemble) on the same embeddings, strengthening the evidence in Figure 3. Var-T-JEPA leads on Spearman correlation with the simulated uncertainty score and on AUROC for detecting high-ambiguity samples; the only exception is AURC, where MC dropout and deep ensembles perform best.
>
> |Dataset|Method|Spearman|AUROC high ambiguity|AURC|
> |-|-|-|-|-|
> |**SIM**|Var-T-JEPA (latent uncertainty)|**0.640**|**0.805**|0.209|
> ||Probe entropy|0.339|0.692|0.272|
> ||Probe MSP|0.313|0.668|0.267|
> ||MC dropout|0.560|0.599|0.013|
> ||Deep ensemble|0.586|0.612|**0.011**|
> |**MNIST**|Var-T-JEPA (latent uncertainty)|**0.949**|**0.986**|0.123|
> ||Probe entropy|0.476|0.773|0.054|
> ||Probe MSP|0.462|0.764|0.054|
> ||MC dropout|0.379|0.763|**0.025**|
> ||Deep ensemble|0.444|0.782|0.028|
>
> **6. Is the contribution incremental VAE machinery?**
>
> We respectfully disagree: the distinction between JEPA and generative models is not structural. We show that JEPA is a deterministic special case of a coupled latent-variable model with a learned conditional prior, unifying predictive and generative self-supervision and opening JEPA to well-established generative approaches (richer priors, hierarchical latents) rather than a separate paradigm, while yielding practical benefits: improved robustness and principled uncertainty estimation.

---

> > ### Author Rebuttal · Reviewer_18NP · 2026-04-05
> >
> > I am marking my concerns as fully resolved based on the authors' comprehensive rebuttal and the inclusion of new empirical data that directly addresses my initial critiques.
> >
> > Empirical Validation of Uncertainty: The authors provided a new comparative study against established baselines (MC Dropout, Deep Ensembles). The results demonstrate that 'Var-T-JEPA outperforms these methods' in Spearman correlation and high-ambiguity detection, providing the "actionable signal" evidence I requested.
> > Justification of Tabular Domain: The response clarified that using tabular data was a strategic choice to 'isolate the variational objective' from the confounding architectural variables (masking, scaling) found in vision/video. This frames the work as a fundamental contribution to SSL theory.
> > Computational Transparency: The new benchmarking data confirms that while memory overhead increases, wall-clock training time remains comparable to T-JEPA, alleviating concerns regarding the practical cost of the added VAE machinery.
> > Anti-Collapse Diagnostics: The authors provided clear evidence that the full ELBO objective prevents representation collapse on $s_y$. Their analysis of "Variant G" proves that the variational framework replaces the need for ad-hoc heuristics with principled regularization.
> >
> > By unifying JEPA and generative modeling into a single probabilistic framework, the authors have successfully demonstrated both the theoretical significance and the practical robustness of their approach. I have adjusted my score to reflect this resolution.

---

### Official Review · Reviewer_d64N · 2026-03-10

**Soundness:** 2
**Presentation:** 3
**Significance:** 2
**Originality:** 3
**Overall Recommendation:** 3
**Confidence:** 3

**Summary:**

This paper central objective is to recast JEPA in a probabilistic way and to introduce Var-JEPA, where the predictor is viewed as a learned conditional prior and the model is trained with one ELBO objective. The paper argues that JEPA should not be seen as fully separate from generative latent variable models, and that a probabilistic view gives a more principled way to regularize the latent space.

**Compliance With Llm Reviewing Policy:**

Affirmed.

**Key Questions For Authors:**

Why is the probabilistic framing necessary in practice? The paper argues that it gives a principled way to avoid collapse, model uncertainty, and connect JEPA to generative modeling. However, it is still unclear what concrete limitation of deterministic JEPA cannot be handled by existing deterministic methods and regularizers.

The paper states that the ELBO offers a principled way to avoid collapse, but the ablations show that removing key terms can still lead to collapse or severe failure. Can the authors make this claim more precise?

**Limitations:**

Please see the key questions and weaknesses.

**Strengths And Weaknesses:**

The main strength of the paper is the idea itself, where the probabilistic view of JEPA is clear and interesting. The synthetic study is also useful. The ablations show what happens when KL terms or reconstruction terms are removed.

The main weakness is that the empirical scope is limited. The paper makes broad claims about JEPA and about bridging predictive and generative self-supervised learning, but the experiments are only on tabular data. Another weakness is that the practical gain is not always strong. The method improves over T-JEPA, but it is often only competitive with strong baselines that use raw features. So the paper makes a stronger case for conceptual value than for clear practical advantage. A final concern is that the paper does not fully show why the probabilistic view is needed in practice. It explains what the probabilistic view can provide, but it is still not fully clear what concrete problem cannot be handled by deterministic JEPA with existing regularization methods. This affects both originality and significance.

---

> ### Author Rebuttal · Authors · 2026-03-29
>
> Dear Reviewer,
>
> we appreciate your thorough review and are happy to hear that you find our paper's idea interesting. Before addressing your questions and concerns, we would like to briefly restate our paper's main contribution, as it may be easily misinterpreted:
> ___
>
> ***Why this fundamental reinterpretation of JEPA matters***
>
> *The central claim is not merely that one can add stochasticity to JEPA, but that JEPA's predictor-encoder structure already corresponds to specific components of a unified latent-variable model. Standard JEPA omits the decoders and explicit distributional regularization – LeJEPA later added the latter via SIGReg (which approximates what the ELBO's KL terms enforce), consistent with what Var-JEPA derives from first principles. This matters because Var-JEPA unifies all these components, replacing a collection of separate heuristics with a formulation in which prediction, regularization, and uncertainty arise from the same objective. In our study, once the same predictive backbone is placed inside that principled latent-variable model, the deterministic formulation, i.e. JEPA, no longer looks like the more compelling one: the variational version is richer conceptually, more robust empirically, and additionally provides probabilistically meaningful uncertainty, which in turn makes JEPA look less like a fundamentally separate paradigm and more like a restricted deterministic special case. We therefore believe our contribution is relevant beyond a single new model variant, and that ICML should showcase not only new model architectures and variants, but also fundamental shifts in perspective that clarify what an existing line of work is really doing.*
> ___
>
> **1. Why is the probabilistic framing necessary in practice?**
>
> We see three concrete practical limitations of deterministic JEPA that are addressed by the probabilistic view in our work. First, collapse is not reliably prevented in practice by heuristic stabilizers alone: Table 1 and Figure 2 show that incomplete objectives can collapse or become unstable, whereas the complete ELBO keeps the latent variables well behaved. Second, deterministic JEPA does not provide a native per-sample uncertainty signal, while Var-JEPA yields one directly through its latent distributions; this is already visible in Figure 3 and is further strengthened by the additional uncertainty-baseline comparison included in our reply (point 5) to Reviewer `18NP`. Third, the variational perspective exposes a unified latent-variable interpretation of JEPA, so the predictor, prior geometry, and regularization are derived from a single formulation rather than introduced separately.
>
> **2. Limited empirical scope**
>
> Our decision to focus on tabular data in this initial work was deliberate rather than incidental: it lets us test the variational reformulation itself in a controlled setting where collapse, uncertainty, and latent behavior are directly interpretable. This is precisely what the simulation study and ablations in Table 1 are designed to show. In contrast, a quick port to I-JEPA or V-JEPA would inherit design choices tuned for deterministic objectives, making it much harder to isolate whether improvements come from the variational objective or from unrelated large-scale engineering choices.
>
> **3. Practical gain versus strong raw-feature baselines**
>
> Our paper makes a strong case for conceptual value and robustness of JEPA rather than for universal dominance over strong supervised and unsupervised baselines. We think this is the correct interpretation of the results. Table 2 shows that on datasets where T-JEPA is already stable and the context-to-target mapping is relatively deterministic, the gap is small. However, T-JEPA is considerably less robust across settings, while Var-T-JEPA is consistently more stable and remains competitive with strong raw-feature baselines. Figure 3 further shows that the variational model provides an actionable uncertainty signal in controlled settings. We therefore view robustness, principled regularization, and uncertainty estimation as the main practical benefits of the variational formulation, rather than claiming a uniform accuracy gain in all regimes.
>
> **4. How should the claim about ELBO-based collapse prevention be interpreted?**
>
> Our claim is that the complete ELBO provides principled regularization, not that arbitrary subsets of its terms do. Table 1 and Figure 2 are important precisely because they show that each component has a clear role: reconstruction terms keep the latents informative, while KL terms constrain latent geometry and prevent pathological solutions. Removing these components does not contradict the argument; it clarifies which parts of the unified objective are responsible for the observed stability.

---

### Official Review · Reviewer_e7io · 2026-03-12

**Soundness:** 3
**Presentation:** 3
**Significance:** 3
**Originality:** 3
**Overall Recommendation:** 4
**Confidence:** 3

**Summary:**

This paper proposes Var-JEPA, a variational reformulation of the Joint-Embedding Predictive Architecture (JEPA) by grounding it in a coupled latent-variable model and deriving an ELBO objective. The authors argue that the supposed divide between JEPA and generative modeling is mostly a matter of framing rather than a fundamental structural difference. A tabular instantiation (Var-T-JEPA) is presented and evaluated against T-JEPA and raw-feature baselines across several datasets. The method also enables uncertainty quantification via posterior covariances, which is used for selective prediction.

**Compliance With Llm Reviewing Policy:**

Affirmed.

**Key Questions For Authors:**

see weakness

**Limitations:**

yes

**Strengths And Weaknesses:**

Strengths:

The theoretical connection between JEPA and variational inference is well-motivated and the derivation is clean — this is a genuinely useful conceptual contribution that clarifies a longstanding ambiguity in the JEPA literature.

The collapse prevention mechanism emerging naturally from the ELBO, without needing EMA or ad-hoc regularizers, is an elegant property and well-supported by the ablation results in Table 1.

The uncertainty quantification capability is practically useful and the risk-coverage curves on MNIST and SIM provide decent evidence that the learned uncertainty is meaningful.


Weaknesses:

Experiments are limited to tabular data only, which is a relatively low-stakes domain for JEPA-style methods , the paper's claims about bridging predictive and generative SSL would be far more convincing with even preliminary results on image data like CIFAR or ImageNet subsets.

The gains over T-JEPA on several real-world datasets are quite marginal (e.g., AD, CC, BM), and on some datasets like CO and SIM the variance is high enough that its hard to draw strong conclusions about consistent improvement.

The paper introduces multiple hyperparameters (five separate loss weights plus annealing schedules for each KL term) and the sensitivity analysis in Appendix B only covers the Adult dataset , its not clear how sensitive these settings are across other datasets.

---

> ### Author Rebuttal · Authors · 2026-03-29
>
> Dear Reviewer,
>
> thank you for your positive feedback and for recognizing the conceptual contribution. At its core, the paper reinterprets standard JEPA as a deterministic special case of a broader variational latent-variable framework, which we find to be conceptually richer and empirically more robust; below we address your questions and concerns.
>
> **1. Restriction to tabular data**
>
> The choice of tabular data is deliberate: it lets us evaluate the variational reformulation in a controlled setting where collapse behavior, latent geometry, and uncertainty can be diagnosed directly from the objective, as in Table 1, Table 2, and Figure 3. In contrast, image/video JEPA systems introduce many additional factors (scale, masking policy, tokenization, optimizer details) that can dominate behavior and make objective-level attribution much less clear. Tabular JEPA-style modeling is also active in other application domains (e.g., Gene-JEPA and Cell-JEPA in single-cell biology), so this setting is not merely a toy endpoint. That said, vision and video are important next steps; Appendix A.4 discusses how the same variational formulation transfers to vision/video architectures. Video also presents an additional opportunity: V-JEPA predicts next-frame target embeddings from context embeddings via a deterministic latent predictor. Casting this as an explicit learned conditional distribution – as Var-JEPA does – opens the full design space of probabilistic latent dynamic models, including normalizing flows and diffusion models in latent space, enabling diverse future generation and calibrated uncertainty.
>
> **2. Marginal gains over T-JEPA on some datasets**
>
> The non-uniform gains are consistent with our central claim rather than a contradiction of it. Table 2 shows that when the context-target mapping is relatively deterministic and T-JEPA already remains stable, large margins are naturally less likely. The stronger pattern is robustness: T-JEPA is considerably less stable across datasets, while Var-T-JEPA is consistently more stable and remains competitive with strong raw-feature baselines; Figure 3 further shows that the variational model provides a useful uncertainty signal in controlled ambiguity setting.
>
> This directly illustrates the central point in the paper that we would like to emphasize here again: ***the broader variational formulation – of which current JEPA is only a restricted special case – provides a cleaner objective, better-behaved training, and principled uncertainty, instead of relying on separately engineered stabilizers; we believe this realization is directly useful for the large community working on this active line of research.***
>
> **3. Hyperparameter sensitivity beyond Adult**
>
> To address this concern, we repeated the Appendix B architecture/optimization sensitivity analysis on both Bank Marketing (BM) and Credit Card Default (CC), using the same ablation family as Table 5 and reporting mean downstream test accuracy across 5 runs. The pattern is consistent with Adult: performance is broadly stable across most architecture changes, and selective evaluation typically improves accuracy as uncertain samples are filtered out.
>
> ||Shallow|Low pred.-FF|Narrow|High pred.-FF|Deep|Low LR|Low batch size|
> |-|-|-|-|-|-|-|-|
> |**Bank Marketing (BM)**||||||||
> |MLP + Var-T-JEPA|0.900|0.900|0.898|0.900|0.898|0.898|0.899|
> |MLP + Var-T-JEPA (20%)|0.912|0.908|0.912|0.908|0.907|0.895|0.911|
> |MLP + Var-T-JEPA (50%)|0.917|0.921|0.924|0.921|0.917|0.887|0.918|
> |ResNet + Var-T-JEPA|0.898|0.900|0.898|0.900|0.899|0.899|0.898|
> |ResNet + Var-T-JEPA (20%)|0.911|0.910|0.914|0.910|0.909|0.897|0.909|
> |ResNet + Var-T-JEPA (50%)|0.918|0.920|0.926|0.920|0.920|0.890|0.917|
> |**Credit Card Default (CC)**||||||||
> |MLP + Var-T-JEPA|0.792|0.818|0.808|0.818|0.807|0.814|0.774|
> |MLP + Var-T-JEPA (20%)|0.814|0.835|0.833|0.835|0.818|0.830|0.849|
> |MLP + Var-T-JEPA (50%)|0.829|0.841|0.845|0.841|0.848|0.845|0.853|
> |ResNet + Var-T-JEPA|0.814|0.813|0.808|0.813|0.801|0.816|0.226|
> |ResNet + Var-T-JEPA (20%)|0.822|0.833|0.834|0.833|0.811|0.831|0.151|
> |ResNet + Var-T-JEPA (50%)|0.832|0.841|0.848|0.841|0.835|0.846|0.147|
>
> We also repeated the Appendix loss-term (“$\alpha$”) end-weight sensitivity study from Table 4 on Bank Marketing, using the same seven settings for $(\alpha_{s_x}^{\text{KL}}, \alpha_{s_y}^{\text{KL}}, \alpha^{\text{rec}}, \alpha^{\text{gen}})$ as in the paper. Again, we find that Var-T-JEPA is robust across different loss-term weights.
>
> ||Base|Low KL|High KL|Low recon.|High recon.|High KL + low recon.|Low KL + high recon.|
> |-|-|-|-|-|-|-|-|
> |MLP + Var-T-JEPA|0.900|0.900|0.888|0.899|0.884|0.898|0.899|
> |MLP + Var-T-JEPA (20%)|0.908|0.903|0.908|0.905|0.882|0.907|0.900|
> |MLP + Var-T-JEPA (50%)|0.921|0.891|0.915|0.902|0.877|0.914|0.910|
> |ResNet + Var-T-JEPA|0.900|0.899|0.889|0.898|0.886|0.898|0.899|
> |ResNet + Var-T-JEPA (20%)|0.910|0.901|0.908|0.903|0.883|0.906|0.900|
> |ResNet + Var-T-JEPA (50%)|0.920|0.890|0.915|0.902|0.876|0.913|0.908|

---

### Official Review · Reviewer_mnen · 2026-03-13

**Soundness:** 3
**Presentation:** 3
**Significance:** 3
**Originality:** 3
**Overall Recommendation:** 4
**Confidence:** 3

**Summary:**

This paper connects the Joint-Embedding Predictive Architecture (JEPA), a meta-architecture for self-supervised learning (SSL), and probabilistic generative modeling by reinterpreting JEPA through the lens of variational inference. The authors demonstrate that the canonical JEPA design is formally connected to the variational posteriors, decoders, and learned conditional priors of coupled variational autoencoders (VAEs). By optimizing the Evidence Lower Bound (ELBO) of the joint log-likelihood, the proposed Variational JEPA (Var-JEPA) makes the latent generative structure explicit, effectively replacing ad-hoc anti-collapse heuristics with the regularization terms in the ELBO. This framework enables robust uncertainty quantification in the latent space and induces representation isotropy via the Kullback-Leibler divergence term without requiring explicit distribution regularization such as Sketched Isotropic Gaussian Regularization (SIGReg). Empirical results on tabular data (Var-T-JEPA) asserts that this variational approach improves representation learning performance over the deterministic baseline T-JEPA.

**Compliance With Llm Reviewing Policy:**

Affirmed.

**Key Questions For Authors:**

1. Can this generative framework learn robust representations without manual data augmentation, and how can this capability be empirically validated?

2. What semantic information does $z$ encode, and what empirical evidence (e.g., probing or latent traversal) proves it is not mere stochastic noise?

3.To verify the framework’s generalizability, could Var-JEPA be effectively extended to other modalities such as I-JEPA [1] and V-JEPA [2]?

[1] I-JEPA: Self-Supervised Learning from Images with a Joint-Embedding Predictive Architecture
[2] V-JEPA: Revisiting Feature Prediction for Learning Visual Representations from Video

**Limitations:**

Insufficient empirical substantiation of the framework's internal probabilistic dynamics and cross-modal scalability.

**Strengths And Weaknesses:**

## Strengths

1. It is interesting to see that JEPA not only resembles generative architecture but also can be interpreted as a genuine generative model. This paper demonstrates that the structural design of JEPA is inherently generative, and that its perceived separation from generative modeling is merely a result of not being explicitly formulated in probabilistic terms. Consequently, the JEPA framework is gracefully embraced within the framework of variational inference.

2. The most significant practical benefit would be getting rid of designing task-specific regularizers. Regularlzation is built-in within the VAE framework, and, if necessary, the vast literature on VAE can be combined to extend JEPA. Additionally, this framework can be used to quantify uncertainty of latent variable mapping.

3. The simulation results support the theoretical findings that $s_x$ is regularized toward $N(0, I)$ but $s_y$ is toward the different target distribution.

## Weaknesses

1. Despite the intriguing theory, the empirical studies are limited to very small datasets (as large as MNIST) and restricted to comparison to T-JEPA on tabular data. Larger and more diverse datasets are desirable to compare the probabilistic and non-probabilistic implementations of JEPA.

2. More investigation of quantitative evidence is recommended regarding the internal behavior of the probabilistic framework, such as the potential posterior collapse of $z$ ($\text{KL}(q_\phi(z|s_x) \| p(z)) \approx 0$) or the numerical results of the ELBO surgery decomposition.


## Minor comments

1.  The readability of the text in the figures and tables is poor.

---

> ### Author Rebuttal · Authors · 2026-03-29
>
> Dear Reviewer,
>
> thank you for your thoughtful assessment. We are pleased that you found our paper interesting.  Below, we explicitly address your questions and concerns.
>
> **1. Robust representations without manual augmentation**
>
> Like JEPA, our framework does not rely on manual data augmentation. In the tabular setting, both Var-T-JEPA and T-JEPA are trained with the same feature-level masking strategy. However, in Var-T-JEPA, the ELBO couples prediction with reconstruction and KL regularization in a single probabilistic objective, which keeps latent geometry well-behaved and discourages degenerate solutions. In T-JEPA, by contrast, stabilization is handled more indirectly through separate heuristic design choices, so collapse-like failures can still occur in difficult settings. Empirically, this is reflected in Table 2: Var-T-JEPA appears to be markedly more stable than T-JEPA across datasets.
>
> This reinforces our central point: *Var-JEPA is not simply an improved JEPA variant, but a paradigm-level reinterpretation of JEPA itself – what is often treated as a separate non-generative approach appears instead as a restricted deterministic limit of a unified latent-variable framework; given the current prominence of JEPA-style methods, we believe this is a timely and useful understanding for many researchers working on self-supervised learning.*
>
> **2. What quantitative evidence do we have for internal probabilistic behavior, and what does $z$ encode?**
>
> The paper gives two pieces of evidence for internal probabilistic behavior in general. First, Table 1 and Figure 2 show that the probabilistic components remain active and that removing key ELBO terms leads to pathological behavior. Second, Figure 3 shows that the model's latent uncertainty signal is aligned with simulated ambiguity on the controlled MNIST and SIM datasets. To answer the $z$-specific question more directly, we extend the paper's Section 4.1 (Simulation Study) with additional diagnostics focused on the auxiliary latent $z$, and explicitly explore low ($\sigma_z = 0.1$) and high simulated residual uncertainty ($\sigma_z = 1.0$). In the low-uncertainty regime, the full ELBO gives substantially better covariance matching than the $\alpha_z^{\text{KL}}=0$ ablation, while across both regimes it keeps the posterior over $z$ prior-consistent (very small $\mathrm{KL_z}$, posterior std close to $1$). By contrast, removing $\mathrm{KL_z}$ drives $q_\phi(z \mid s_x)$ into a clearly pathological regime with very large $\mathrm{KL_z}$ and near-collapsed posterior variance.
>
> | Residual uncertainty | Case | KL$_z$ | Posterior std(z) | Residual-cov Fro. error | Residual-cov trace ratio |
> | --- | --- | ---: | ---: | ---: | ---: |
> | Low | ELBO | **2.66e-4 ± 1.53e-5** | **0.998 ± 2.53e-4** | **1.157 ± 0.117** | **1.769 ± 0.070** |
> | Low | ELBO without $\mathrm{KL_z}$ | 56.797 ± 2.715 | 0.012 ± 1.07e-3 | 2.577 ± 0.205 | 3.434 ± 0.174 |
> | High | ELBO | **2.74e-4 ± 1.54e-5** | **0.998 ± 3.77e-4** | 6.646 ± 0.841 | 0.522 ± 0.038 |
> | High | ELBO without $\mathrm{KL_z}$ | 57.948 ± 5.626 | 0.015 ± 2.78e-3 | **6.064 ± 0.820** | **0.742 ± 0.049** |
>
> These additional diagnostics suggest that $z$ captures residual predictive uncertainty, that is, variability in the target latent that is not explained by the context alone. The key point is not a large magnitude of $z$, but whether the auxiliary latent remains well-regularized and functionally useful under the full ELBO; in contrast, removing $\mathrm{KL_z}$ drives the posterior into a pathological regime. $z$ can also be omitted, with this choice governed by standard model selection criteria, consistent with the fact that JEPA formulations may or may not include an explicit stochastic latent. For an additional baseline comparison on uncertainty quantification, we refer to our reply to Reviewer `18NP` (point 5).
>
> **3. Could Var-JEPA extend to I-JEPA and V-JEPA?**
>
> Yes. As discussed in Appendix A.4, the core architectural pattern of JEPA already matches the latent-variable factorization we derive, so the variational objective can in principle replace the deterministic one in image or video settings as well. An especially interesting case arises for video: V-JEPA predicts next-frame target embeddings from context deterministically. Once cast as an explicit conditional distribution in the Var-JEPA framework, this opens the design space of probabilistic latent dynamic models – including flows and diffusion models over latents – enabling diverse generation and calibrated uncertainty over future content. In this paper, however, we deliberately focused on tabular data because it lets us validate the variational reformulation in a more controlled setting, without entangling that objective-level question with the many architecture and training choices that are already built into large-scale visual JEPA systems.

---

> > ### Author Rebuttal · Reviewer_mnen · 2026-04-03
> >
> > The rebuttal properly addressed my concerns.

---

### Decision · Program_Chairs · 2026-04-30

**Decision:**

Accept (regular)

**Comment:**

The manuscript introduces a probabilistic variational version of the JEPA architecture.

The reviewers appreciated the elegant theoretical justification and its practical benefits (avoiding heuristics and incorporating uncertainty quantification). The however also judged the methodological novelty to be somewhat limited, the variational implementation being a relatively straightforward extension of the original architecture. Experiments were also considered relatively limited (strong focus on tabular data).

Overall, I think the manuscript has benefits, for the community investigating JEPA-style architectures, that outweight the weaknesses and I recommend acceptance if possible.